# REVISITING ZEROTH-ORDER OPTIMIZATION: MINIMUM-VARIANCE TWO-POINT ESTIMATORS AND DIRECTIONALLY ALIGNED PERTURBATIONS

**Shaocong Ma and Heng Huang**\*
Department of Computer Science
University of Maryland
College Park, MD 20742, USA
{scma0908, heng}@umd.edu

## ABSTRACT

In this paper, we explore the two-point zeroth-order gradient estimator and identify the distribution of random perturbations that minimizes the estimator's asymptotic variance as the perturbation stepsize tends to zero. We formulate it as a constrained functional optimization problem over the space of perturbation distributions. Our findings reveal that such desired perturbations can align directionally with the true gradient, instead of maintaining a fixed length. While existing research has largely focused on fixed-length perturbations, the potential advantages of directional alignment have been overlooked. To address this gap, we delve into the theoretical and empirical properties of the directionally aligned perturbation (DAP) scheme, which adaptively offers higher accuracy along critical directions. Additionally, we provide a convergence analysis for stochastic gradient descent using $\delta$-unbiased random perturbations, extending existing complexity bounds to a wider range of perturbations. Through empirical evaluations on both synthetic problems and practical tasks, we demonstrate that DAPs outperform traditional methods under specific conditions.

## 1 INTRODUCTION

*Zeroth-order optimization (ZOO)* has emerged as a crucial paradigm in machine learning and optimization, particularly in scenarios where gradient information is unavailable or prohibitively expensive to compute. This approach has garnered significant attention in diverse applications, including black-box adversarial attacks on machine learning models (Papernot et al., 2017; Chen et al., 2017; Kurakin et al., 2016; Cai et al., 2021), physical-informed neural networks with external black-box PDE solvers (Shen et al., 2024; Ma et al., 2025), memory-efficient fine-tuning of large language models (Malladi et al., 2023; Zhang et al., 2024; Gautam et al., 2024), and reinforcement learning (Choromanski et al., 2018; Lei et al., 2022). In recent years, the memory-efficient consideration motivates the one-bit approach (Cai et al., 2022a) and the study on the sparsity of the gradient (Cai et al., 2022b). The randomized method (Akhavan et al., 2022) has also emerged as a critical direction.

In this paper, we consider the following unconstrained stochastic optimization problem:

$$\min_{x \in \mathbb{R}^d} f(x) := \mathbb{E}_{\xi \sim \Xi} f(x; \xi), \tag{1}$$

where the random data $\xi$ is sampled from the underlying distribution $\Xi$, and the objective function $f(x)$ is defined as the expectation of the smooth individual loss function $f(x; \xi)$. While traditional first-order methods utilize the stochastic gradient $\nabla f(x; \xi)$ to update parameters, zeroth-order optimization relies solely on function evaluations. A common approach in this context is to use the standard two-point estimator to approximate the gradient, given by:

$$\hat{\nabla} f(x; \xi, v) := \frac{1}{\mu} \left[ f(x + \mu v; \xi) - f(x; \xi) \right] v, \tag{2}$$

---

\*This work was partially supported by NSF IIS 2347592, 2348169, DBI 2405416, CCF 2348306, CNS 2347617.

where $\mu \in \mathbb{R}_+$ is the perturbation stepsize and $v \in \mathbb{R}^d$ is a random vector, typically drawn from a uniform distribution on the sphere or a standard Gaussian distribution.

The theoretical analysis of zeroth-order optimization methods has been extensively studied in the existing literature. Numerous studies have provided valuable insights into the convergence properties and performance guarantees of these methods under various conditions (Ghadimi & Lan, 2013; Duchi et al., 2015; Ji et al., 2019; Sahu et al., 2019; Coope & Tappenden, 2020; Kozak et al., 2023; Rando et al., 2024a;b). However, when studying algorithms like SGD in the zeroth-order setting, these analyses typically consider specific types of random perturbations, such as Gaussian or uniform distributions. While this approach has yielded important theoretical results, it often lacks a comprehensive explanation for why these particular random perturbations are optimal with respect to the variance of gradient estimator, which suggests the need for a more comprehensive framework that can accommodate a broader class of random perturbations. This observation leads us to the central question in this paper:

> *Q1: How can we determine the class of distributions of random perturbation in a zeroth-order estimator to minimize its variance?*

**Contribution 1**: To address this central question, we develop a novel approach based on solving the following constrained functional optimization problem:

$$\min_V \mathbb{E}_{v \sim V} \|\hat{\nabla} f(x; \xi, v) - \nabla f(x; \xi, v)\|^2$$
$$\text{s.t. } \mathbb{E}_{v \sim V} v v^\top = \delta I_d,$$

where $\hat{\nabla} f(x; \xi, v)$ is the two-point gradient estimator for approximating the stochastic gradient $\nabla f(x; \xi)$ as defined in Eq. (2), and $\delta > 0$ is a given scaling constant. This optimization problem is formulated over a functional space encompassing all probability distributions over $\mathbb{R}^d$, subject to a linear constraint. The constraint follows the existing zeroth-order optimization literature (Kozak et al., 2023; Rando et al., 2024b), which ensures that the direction of the two-point gradient estimator is the same as the true gradient as the perturbation step $\mu$ is sufficiently small. Moreover, this constraint is linear in the distribution $V$ when treating it as an integral with respect to a general measure $dV$. The solution to this optimization problem reveals a broader condition such that the two-point gradient estimator achieves the minimum variance: either (a) the perturbation vector $v$ should have a fixed length, or (b) the inner product between $v$ and the true gradient $\nabla f(x; \xi)$ should have a fixed magnitude. These insights extend beyond a specific type of distribution such as uniform distributions over the sphere, enabling us to characterize a more general class of effective perturbations. These findings naturally lead us to our second question:

> *Q2: Can we leverage these theoretical insights to design novel random perturbation schemes that outperform existing methods?*

**Contribution 2**: Existing literature reveals that most perturbation methods focus primarily on ensuring a fixed length for the perturbation vector $v$ (for more details, we include a brief discussion in Appendix A.2). In contrast, we propose a novel scheme based on our second condition: choosing the random perturbation $v$ such that

$$\left(\nabla f(x; \xi, v)^\top v\right)^2 = \delta \|\nabla f(x; \xi, v)\|^2$$

where $\delta > 0$ is a given constant. In practice, we replace the true stochastic gradient $\nabla f(x; \xi, v)$ with its estimation $\hat{\nabla} f(x; \xi, v)$. This new random perturbation offers several advantages: (1) It extends the minimum-variance random perturbation design, which is particularly drawn by the theoretical interest. (2) When certain components of the gradient are large, our proposed method exhibits strong anisotropic behavior and presents higher accuracy in these directions. This characteristic indicates that our approach is more focused on effective dimensions, potentially leading to more efficient optimization in high-dimensional spaces with sparse gradients.

**Contribution 3**: Lastly, to validate the empirical performance of our proposed perturbation scheme, we conduct extensive experiments across multiple domains to demonstrate the effectiveness of our proposed method. On synthetic optimization problems, we show that our approach achieves significantly higher accuracy in gradient estimation compared to standard methods. In language model fine-tuning tasks, we apply our method to optimize the OPT-1.3b model on the SST-2 dataset, demonstrating faster convergence and higher final accuracy relative to existing zeroth-order approaches.

## 1.1 PAPER STRUCTURE

The remainder of this paper is organized as follows: In Section 2, we analyze the two-point gradient estimator and derive the sufficient conditions for minimum-variance random perturbations. This analysis reveals two distinct classes of perturbations which minimize the asymptotic variance (as the perturbation step $\mu \to 0$) of two-point gradient estimation: fixed-length perturbations and directionally aligned perturbations (DAPs). In Section 3, we present the convergence analysis for SGD algorithm using $\delta$-unbiased random perturbations. We establish the complexity that matches the common dependence on the dimension $d$ while extends to a broader class of perturbations, including both traditional fixed-length methods and our proposed DAPs. In Section 4, we introduce the DAP in detail. We examine its unique properties, particularly their anisotropic behavior and ability to adapt to gradient magnitudes across different dimensions. We also present a practical algorithm for implementing DAPs. In Section 5, we provide extensive experimental validation of our theoretical findings. We evaluate DAPs against traditional perturbation methods on both synthetic optimization problems and practical machine learning tasks.

## 2 THE DERIVATION OF MINIMUM-VARIANCE RANDOM PERTURBATIONS

In this section, we consider the two-point gradient estimator for approximating the gradient $\nabla f(x)$ and seek to derive the minimum-variance random perturbation strategy.

In gradient-based optimization, the descent direction is the opposite of the gradient vector. This naturally leads to the following unbiased assumption up to a scaling constant $\delta$ which ensures that the direction of estimated gradient is aligned with the true gradient:

**Assumption 2.1** ($\delta$-Unbiasedness). *Let the two-point gradient estimator for estimating the stochastic gradient $\nabla f(x; \xi)$ be $\hat{\nabla} f(x; \xi, v) := \frac{1}{\mu} \left[ f(x + \mu v; \xi) - f(x; \xi) \right] v$. The distribution $V$ satisfies the $\delta$-unbiasedness if $\mathbb{E} v v^\top = \delta I_d$, where $I_d$ is the identity matrix with the dimension $d$.*

It should be noted that the concept of unbiasedness here is in the asymptotic sense, which holds when $\mu \to 0$. This assumption is commonly satisfied in existing zeroth-order optimization literature. Many popular random perturbation distributions satisfy this assumption, including Gaussian perturbation (Ghadimi & Lan, 2013; Duchi et al., 2015; Nesterov & Spokoiny, 2017), uniform distribution over a sphere (Lin et al., 2022; Duchi et al., 2015), random coordinate/direction sampling (Zhang et al., 2020; Coope & Tappenden, 2020; Kozak et al., 2023), and Rademacher distribution (Spall, 1987; 1992). Building on this assumption, our next goal is to characterize the accuracy of $\hat{\nabla} f(x; \xi, v)$ in approximating the true gradient $\nabla f(x; \xi, v)$, which may lead to insights applicable across various perturbation schemes.

Let $V$ be a distribution such that for any $v \sim V$, $\mathbb{E}[v v^\top] = \delta I_d$, where $I_d$ denotes the $d \times d$ identity matrix. The approximation error of the two-point zeroth-order gradient estimator for estimating $\nabla f(x)$ can be represented as:

$$\mathbb{E}_{v \sim V} \|\hat{\nabla} f(x; v) - \nabla f(x)\|^2 = \mathbb{E}_{v \sim V} \left\| \frac{1}{\mu} [f(x + \mu v) - f(x)] v - \nabla f(x) \right\|^2$$

$$(\mu \to 0) \quad \overset{(i)}{\approx} \mathbb{E}_{v \sim V} \|(v v^\top - I) \nabla f(x)\|^2$$

$$= \mathbb{E}_{v \sim V} \nabla f(x)^\top (v v^\top)^2 \nabla f(x) + (1 - 2\delta) \|\nabla f(x)\|^2.$$

where (i) applies the Taylor theorem $f(x + \mu v) - f(x) = \mu \langle \nabla f(x), v \rangle + \frac{\mu^2}{2} \langle M_c(v), v \rangle$ (Lemma C.9) and assumes the perturbation step $\mu$ is sufficiently small.

*Remark* (Taylor Approximation Error). We note that the above approximation and the unbiasedness of the two-point gradient estimation require the perturbation step $\mu \to 0$. This requirement is commonly made in practice such as the language model fine-tuning (Malladi et al., 2023) and other theoretical convergence analysis (Duchi et al., 2015). With making additional assumptions such as the $L$-smoothness (Assumption C.1), we can have more accurate upper bound which we will use in our final convergence analysis presented in Theorem 3.1 and Corollary 3.2. More explicitly, if the function $f$ is $L$-smooth, then it must have a $L$-bounded Hessian matrix. Therefore, we obtain

$$\mathbb{E} \|\hat{\nabla} f(x; v) - \nabla f(x)\|^2 \leqslant \nabla f(x)^\top (v v^\top)^2 \nabla f(x) + (1 - 2\delta) \|\nabla f(x)\|^2 + L^2 \mu^2 \mathbb{E} \|v\|^4.$$

We will later bound the first term with respect to $\|\nabla f(x)\|^2$. For the last term, we will make the perturbation step $\mu$ to be sufficiently small to control its magnitude. Due to the page limit, we include more detailed discussions on the accumulative error caused by the gradient approximation and the Taylor approximation in Appendix F.

For notational convenience, we let $a := \nabla f(x)$. Our objective is to find the distribution $V$ that minimizes the above expectation, formalized as:

$$\min_{V} \quad \mathbb{E}_{v \sim V} a^\top (vv^\top)^2 a \tag{3}$$

$$\text{s.t.} \quad \mathbb{E}_{v \sim V} vv^\top = \delta I_d.$$

The optimization problem we have formulated presents two significant challenges: (1) It requires us to optimize over an infinite-dimensional space of probability distributions, a task that demands sophisticated mathematical tools.(2) The constraint $\mathbb{E}_{v \sim V} vv^\top = \delta I_d$ imposes specific second-order moment conditions on the distribution, adding a layer of complexity to our analysis. To address these challenges, we develop a novel analytical approach, described in the following theorem:

**Theorem 2.2.** *Let $v$ be a random vector following the distribution $V$ with $\mathbb{E}_{v \sim V} vv^\top = \delta I_d$ and $a \in \mathbb{R}^d$ be a fixed vector. Then*

$$d\delta^2 \|a\|^2 \leqslant \mathbb{E}_{v \sim V} a^\top (vv^\top)^2 a \leqslant \delta^2 d \|a\|^2 + \frac{\|a\|^2}{2} \rho_V + \frac{\|a\|^2}{2} \sqrt{\rho_V^2 + 4\delta^2 (d-1)\rho_V},$$

*where $\rho_V := \mathbb{E}\|v\|^4 - \delta^2 d^2$. The equality holds if one of the following conditions holds:*

*(a) $\|v\|^2 = d\delta$ holds almost surely.*

*(b) $(a^\top v)^2 = \delta\|a\|^2$ holds almost surely.*

*Remark.* When the equality holds, the two-point gradient estimator defined as Eq. (2) has the minimum variance $\mathbb{E}_{v \sim V} \|\hat{\nabla} f(x; v) - \nabla f(x)\|^2 = (\delta^2 d - 2\delta + 1)\|\nabla f(x)\|^2 + \mathcal{O}(\mu)$.

*Proof.* The details can be found in Appendix B.1. Let $f = (v^\top a)v - \delta a$ and $g = (\|v\|^2 - \delta d)a$. Then we obtain

$$(\mathbb{E}[g^\top f])^2 \overset{(i)}{\leqslant} \mathbb{E}\|f\|^2 \mathbb{E}\|g\|^2,$$

where (i) applies the Cauchy-Schwarz inequality. Then we obatin a quadratic function with respect to the objective function $\mathbb{E}_{v \sim V} a^\top (vv^\top)^2 a$. By analyzing the equality condition of the Cauchy-Schwarz inequality (i.e. $f$ and $g$ are linearly dependent), we obtain the sufficient and necessary condition for achieving the minimum variance. $\qquad\square$

**On the Necessity of Minimum Variance Conditions**  In our previous theorem (Theorem 2.2), we only present the sufficient condition of achieving the asymptotic minimum variance as the perturbation step $\mu$ tends to $0$. A simple counterexample demonstrates that this condition may not be necessary: consider a mixed distribution that takes the DAP with probability $p$ and the uniform perturbation over the sphere with probability $1 - p$. Such a distribution would also achieve minimum variance while satisfying neither condition exclusively. Extending the condition to sufficient and necessary condition would be an interesting but challenging topic. Here we present one potential scenario where we may obtain the necessary and sufficient condition: In the one-dimensional case, assuming the random perturbation satisfying $\mathbb{E}v = 0$ and $\mathbb{E}v^2 = 1$. , the unique distribution of achieving the minimum variance is the Rademacher distribution, which is naturally derived by considering the Taylor expansion. This case may further be extended to $d$-dimension with requiring all entries in the random perturbation to be mutually independent; however, this extension would be out of the scope of our paper and excludes many interesting random distributions. We will still stick to our $\delta$-unbiased perturbations (Assumption 2.1) in the remaining of our manuscript.

This theorem reveals two underexplored insights in zeroth-order optimization: (1) The performance of the two-point gradient estimator is significantly influenced by the fourth-order moment of the perturbation distribution $V$. Choosing a $V$ with an infinite fourth-order moment can result in a poorly performing zeroth-order gradient estimator. This highlights the crucial role of perturbation choice in gradient estimation performance. In the meanwhile, when choosing the perturbation distribution $V$

with the minimum variance, the complexity of SGD with two-point gradient estimation achieves the best known sample complexity, which we discuss further in Section 3. (2) The presented condition naturally leads to two distinct classes of random perturbations that achieve the minimum variance:

(a) **Constant Magnitude Perturbations**: This class of perturbations arises from the Constant Magnitude condition: $\|v\|^2 = d\delta$. This condition gives rise to a diverse range of distributions, including:

- Uniform distribution over a sphere: $v$ is uniformly distributed on the surface of a sphere with radius $d$, i.e., $\|v\|^2 = d$.
- Rademacher distribution: Each entry of $v$ is independently sampled as $v_i = \pm\sqrt{\delta}$ with equal probability, resulting in $\|v\|^2 = d\delta$.
- Random coordinate: $v = \sqrt{d\delta}e_i$, where $e_i$ is a standard basis vector chosen uniformly at random from $e_1, ..., e_d$, ensuring $\|v\|^2 = d\delta$.

They all share the property of having a constant $\|v\|^2$, satisfying the condition (a) in Theorem 2.2. Surprisingly, the widely used Gaussian distribution $v \sim \mathcal{N}(0, I_d)$ has kurtosis $\mathbb{E}[\|v\|^4] = d^2 + 2d > d^2$, which exceeds the minimum variance. Therefore, the Gaussian random perturbation doesn't belong to the class of minimum-variance random perturbations, despite its popularity in many optimization algorithms.

(b) **Directionally Aligned Perturbations (DAPs)**: This class of perturbations stems from the condition: $(a^\top v)^2 = \delta\|a\|^2$. This condition suggests that a perturbation that minimizes the asymptotic variance of two-point gradient estimator as $\mu \to 0$ should be distributed on the surface $a^\top v = \pm\sqrt{\delta}\|a\|$. This class of perturbations, while theoretically promising, remains largely underexplored in the existing literature on zeroth-order optimization. We note that in the practice of zeroth order optimization, this condition can not be imposed like it is, since $\nabla f(x)$ is commonly not available. We provide further discussion of this class of perturbations, including potential sampling stratagies and practical considerations, in Section 4.

In the following sections, we will present the sample complexity of SGD under minimum-variance conditions for the two-point gradient estimator. Specifically, we will examine scenarios where Eq. (3) achieves the minimum value of $d\delta^2\|\nabla f(x;\xi)\|^2$. Subsequently, we will explore the properties of DAPs, which is rarely explored in the existing literature.

## 3 CONVERGENCE OF SGD WITH $\delta$-UNBIASED RANDOM PERTURBATIONS

In existing literature, the approximation error of the gradient estimation $E_{v\sim V}\|\hat{\nabla}f(x;v) - \nabla f(x)\|^2$ is often tailored to a specific type of random perturbation by utilizing the analytical form of the probability densities. By applying the upper bound obtained from Theorem 2.2, we are able to build a more general upper bound which maintains the desired dependence on the dimension $d$. In summary, we analyze the convergence of Stochastic Gradient Descent (SGD) with $\delta$-unbiased random perturbations for the optimization problem defined in Eq. (1) with smooth individual loss functions $f(x;\xi)$, on which we make several standard assumptions including the $L$-smoothness (Assumption C.1) and the strong convexity (Assumption C.2) detailed in Appendix C.1.

To solve the optimization problem presented in Eq. (1), we employ the SGD algorithm. Starting from the initial parameter $x_1$, we repeatedly update the parameter with the update rule

$$x_{t+1} = x_t - \eta\hat{\nabla}f(x_t;\xi_t, v_t) \tag{4}$$

for $t = 1, 2, \ldots, T-1$, where $\xi_t \sim \Xi$ is the data point used at $t$-th update and $\hat{\nabla}f(x_t;\xi_t, v_t)$ is the estimation of the true gradient $\nabla f(x_t;\xi_t)$ with $v_t$ sampled from the distribution $V$ as defined in Eq. (2). For non-convex settings, we monitor $\min_{1\leqslant t\leqslant T}\|\nabla f(x_t)\|^2$, the minimum squared gradient norm of the objective function during training. For strongly convex settings, we track $f(x_t) - f^*$, the function value gap, where $f^* := \inf_{x\in\mathbb{R}^d} f(x)$. Both metrics are standard in smooth optimization literature (Ghadimi & Lan, 2013). We also note that the quantity $\min_{1\leqslant t\leqslant T}\|\nabla f(x_t)\|^2$ used in the non-convex setting is not easy to check in the practice especially in the modern machine learning senario, since the parameter space would be extreamly large.

Now we build the convergence analysis of SGD algorithm with $\delta$-unbiased gradient estimators:

**Theorem 3.1.** *Suppose that Assumption 2.1 and Assumption C.1 are satisfied. Let $\{x_t\}_{t=1}^{T}$ be the SGD dynamic solving Eq. (1) generated by the update rule Eq. (4). If the fourth-order moment of the random perturbation is finite (i.e. $\mathbb{E}_{v \sim V} \|v\|^4 < +\infty$), then*

*(a) If the learning rate $\eta \leqslant \min\{\frac{1}{2L}, \frac{1}{L\sqrt{2T(2\delta^2 d + \rho_V + 2\delta + 1)}}\}$, then*

$$\min_{1 \leqslant t \leqslant T} \mathbb{E}\|\nabla f(x_t)\|^2 \leqslant \frac{(f(x_1) - f^*)}{\delta \eta T} + \frac{2\eta}{\delta}\Big[LB^2(1 + \beta_V) + \mu^2 \alpha_V\Big],$$

*where $c$ is the strong-convexity constant defined in Assumption C.2, $\rho_V := \mathbb{E}\|v\|^4 - \delta^2 d^2$, $\alpha_V := L^3 \mathbb{E}\|v\|^4$, $\beta_V := 2\delta^2 d + \rho_V + 1 - 2\delta$, and $B^2 := 2L(f^* - \mathbb{E}_{\xi \sim \Xi} f_\xi^*)$ with $f_\xi^* := \inf_{x \in \mathbb{R}^d} f(x; \xi)$.*

*(b) If the learning rate $\eta \leqslant \min\{\frac{1}{2L}, \frac{\delta c}{4L^2}\frac{1}{2\delta^2 d + 2\delta + 1 + \rho_V}\}$, and additionally, Assumption C.2 is satisfied, then*

$$\mathbb{E}f(x_T) - f^* \leqslant \big(1 - \frac{c}{2}\delta\eta\big)^{T-1}\big(f(x_1) - f^*\big) + \frac{2}{c\delta}\eta\big(LB^2(1 + \beta_V) + \mu^2 \alpha_V\big),$$

*where $\rho_V$, $\alpha_V$, $\beta_V$, and $B^2$ are as defined in (a).*

*Proof.* The proof directly follows Khaled & Richtárik (2022) and Mishchenko et al. (2020) with additionally bounding the error term $\|\hat{\nabla}f(x_t; \xi_t, v_t) - \nabla f(x_t)\|^2$ using Theorem 2.2. See Appendix C.1 for details. □

This convergence upper bound reveals two important insights that have not been comprehensively studied in existing literature:

(a) *The impact of perturbation magnitude $\delta$*: A small-magnitude perturbation $\delta$ consistently leads to more accurate approximation: According to Theorem 2.2, if we are using the uniform distribution over the sphere with $\delta = 1$, the approximation error $\lim_{\mu \to 0} E_{v \sim V} \|\hat{\nabla}f(x; v) - \nabla f(x)\|^2 = (d - 1)\|\nabla f(x)\|^2$. In the meanwhile, if we are using the uniform distribution over the sphere with $\delta = 1/d$, the approximation error is minimized and achieves $\lim_{\mu \to 0} E_{v \sim V} \|\hat{\nabla}f(x; v) - \nabla f(x)\|^2 = (1 - 1/d)\|\nabla f(x)\|^2$. However, a small magnitude $\delta$ results in a $\frac{1}{\delta}$ scale on the convergence upper bound presented in Theorem 3.1. As a result, tuning the hyper-parameter $\delta$ doesn't change our theoretical complexity analysis; therefore, in the following corollary, we only consider the case where $\delta = 1$.

(b) *The influence of fourth-order moment $\mathbb{E}\|v\|^4$*: The fourth-order moment of the random perturbation $\mathbb{E}\|v\|^4$ significantly impacts the convergence performance of the SGD algorithm, a phenomenon previously identified in the literature such as Duchi et al. (2015). Due to the influence of $\mathbb{E}\|v\|^4$, our convergence analysis cannot guarantee that all $\delta$-unbiased random perturbations can achieve the optimal dependence on dimension $d$ as reported in existing lower bounds (e.g. consider any random distribution with the fourth-order moment $d^{2+c}$ for some $c > 0$). However, as demonstrated in Corollary 3.2, all perturbations that achieve minimum variance will attain this best-known dependence on $d$.

**Corollary 3.2.** *Under the same assumptions as Theorem 3.1, let $V$ achieve the minimum variance. Then*

*(a) Let $\delta = 1$, $\eta = \Theta(\frac{\epsilon}{d})$, and $\mu = \mathcal{O}(\frac{\epsilon}{d})$. Then it requires at most $T \leqslant \mathcal{O}(\frac{d}{\epsilon^2})$ iterations to achieve $\min_{1 \leqslant t \leqslant T} \mathbb{E}\|\nabla f(x_t)\|^2 < \epsilon$.*

*(b) Suppose that Assumption C.2 is satisfied. Let $\delta = 1$, $\eta = \Theta(\frac{\epsilon}{d})$, and $\mu = \mathcal{O}(\frac{\epsilon}{d})$. Then it requires at most $T \leqslant \mathcal{O}(\frac{d}{\epsilon})$ iterations to achieve $\mathbb{E}f(x_T) - f^* < \epsilon$.*

By applying this result, we extend the best-known complexity for zeroth-order SGD for smooth objectives to a much broader class of minimum-variance random perturbations, including the uniform smoothing (Bach & Perchet, 2016), Rademacher distribution (Spall, 1987; 1992), coordinate descent (Cai et al., 2021), random subspace (Kozak et al., 2021), and the general orthogonal perturbations (Kozak et al., 2023). Notably, the results presented in existing literature are either tailored to a specific type of random perturbation, or cannot characterize the convergence of SGD under DAPs.

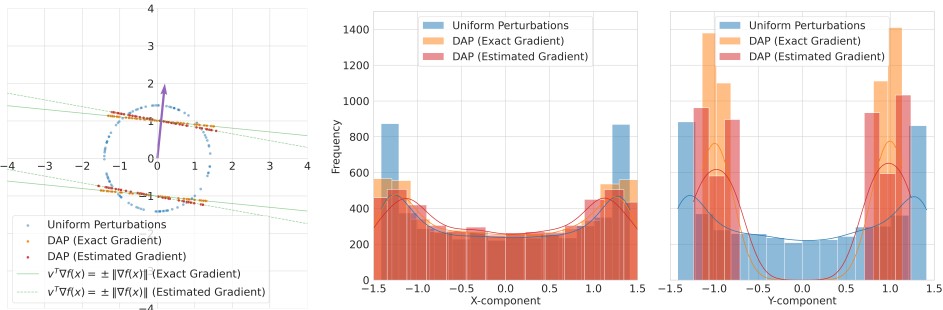

Figure 1: Illustration of the *directional alignment property* of DAP in $d = 2$ with estimating the gradient of $f(x) = x_1^2 + x_2^2$ at $x = \begin{bmatrix} 0.1 & 1 \end{bmatrix}^\top$. We sample 5000 random perturbations from the uniform distribution over the circle $\|x\|^2 = 2$ and DAPs with $a = \nabla f(x) = \begin{bmatrix} 0.2 & 2 \end{bmatrix}^\top$, in which 100 samples are illustrated in the left figure. When projecting DAPs onto different axes, we observe two distinct distributional patterns in the X-component and Y-component, demonstrating DAP's anisotropic nature. In contrast, uniform perturbations maintain same distributions across projections.

## 4 DIRECTIONALLY ALIGNED PERTURBATIONS (DAPs)

In the previous section, we discussed the condition of achieving the minimum variance for approximating the gradient $\nabla f(x)$ using a two-point gradient estimator defined in Eq. (2). We proved that a $\delta$-unbiased random perturbation $V$ achieves the minimum asymptotic variance as $\mu \to 0$ if it has a fixed length or satisfies the following equation:

$$(v^\top \nabla f(x))^2 = \delta \|\nabla f(x)\|^2. \tag{5}$$

We refer to the class of $\delta$-unbiased random perturbations satisfying Eq. (5) as *directionally aligned perturbations (DAP)*. While it has been shown that this class of random perturbations achieves minimum variance, there is limited literature studying their empirical performance. The unknown true gradient could potentially hinder the application of DAP; however, we also recognize several attractive features of DAP in zero-order gradient estimation.

### 4.1 DIRECTIONALLY ALIGNED PROPERTY

The primary characteristic of DAP is its *directional alignment property*. Unlike uniform perturbations or other fixed-magnitude perturbation methods, DAP exhibits anisotropic behavior, meaning its effects vary depending on the direction of projection (as illustrated in Figure 1). This anisotropy is a direct consequence of DAP's design, which inherently leverages information from the objective function's local geometry to perform different perturbation behaviors, which is usually ignored by fixed-magnitude perturbation methods.

As a result of the directional alignment property, when the true gradient has a larger magnitude in a specific direction, DAP adaptively introduces a low-variance perturbation in that direction, potentially leading to higher accuracy. This effect is demonstrated in Figure 2. This adaptive behavior of DAP may reduce noise in estimating gradients, particularly when the gradients are sparse or have varying magnitudes across different dimensions. We further validate this hypothesis and explore its implications in Section 5.

### 4.2 THE SAMPLING STRATEGY AND A PRACTICAL IMPLEMENTATION

While the theoretical properties of DAP are appealing, their practical implementation presents several challenges. For the unknown gradient $\nabla f(x)$, we can always apply a small batch of perturbations to obtain an estimated gradient $\hat{\nabla} f(x)$. However, even with an estimated gradient, sampling from the hyperplane $(v^\top \hat{\nabla} f(x))^2 = \delta \|\hat{\nabla} f(x)\|^2$ satisfying $\mathbb{E} v v^\top = \delta I_d$ remains challenging. This necessitates the design of a specific sampling strategy to address this issue. Moreover, the use of a batch of function evaluations for gradient estimation raises a practical consideration about whether and how to reuse these estimators.

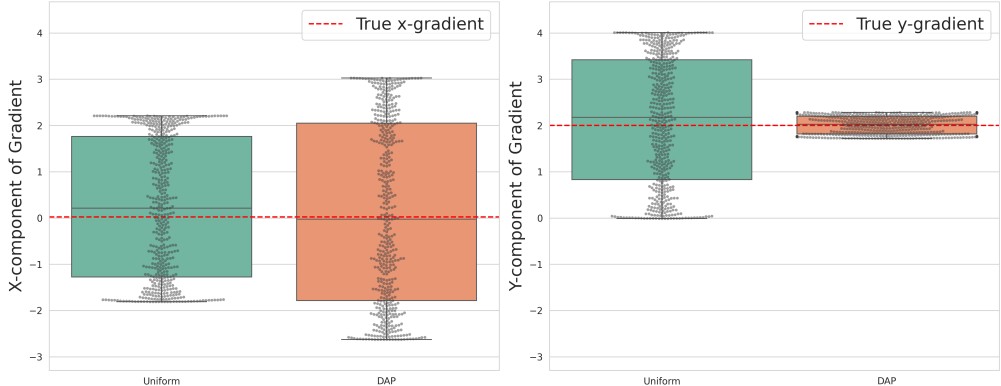

Figure 2: Comparison of gradient estimation performance with estimating the gradient of $f(x) = x_1^2 + x_2^2$ at $x = \begin{bmatrix} 0.1 & 1 \end{bmatrix}^\top$ between Uniform perturbations and DAPs. The DAP exhibits a notably smaller variance in the y-direction where the true gradient has a larger magnitude, compared to the x-direction where the true gradient is close to zero. In contrast, the uniform perturbation method shows similar variance in both directions, regardless of the true gradient's magnitude.

To address these issues, we propose the following practical approach to sample from the DAP distribution. Here, Algorithm 1 describes the approach to generate the random vectors over the plane $(v^\top a)^2 = \delta \|a\|^2$ satisfying $\mathbb{E}vv^\top = \delta I_d$ with a given vector $a \in \mathbb{R}^d$; its correctness is proved in Proposition 4.1. And Algorithm 2 describes the practical zeroth-order gradient estimator using DAPs.

---

**Algorithm 1:** The algorithm for sampling from a hyper-plane

**Input:** The vector $a \in \mathbb{R}^d$

1  Generate a random vector $v_{\text{ini}}$ such that $\mathbb{E}_{v_{\text{ini}} \sim V}[v_{\text{ini}} v_{\text{ini}}^\top] = \delta I_d$;
2  Generate an independent random variable $\xi$ uniformly from the set $\{-1, +1\}$;
3  Project $v_{\text{ini}}$ onto the random plane $\mathcal{P}_\xi = \{v : a^\top v = \xi \sqrt{\delta} \|a\|\}$;

**Output:** The projected random vector $v := \text{Proj}_{\mathcal{P}_\xi}(v_{\text{ini}})$

---

**Algorithm 2:** A practical implementation of gradient estimator using DAPs

**Input:** The dimension $d$, the batch size $b$

```
/* Estimate the gradient     */
```
1  Use $b//2$ uniform perturbations to obtain the gradient estimator $\hat{\nabla} f(x)$;
```
/* Generate DAPs             */
```
2  Use $\hat{\nabla} f(x)$ as the input of Algorithm 1 to obtain $b//2$ DAPs;
3  Use $b//2$ DAPs to obtain another gradient estimator $\tilde{\nabla} f(x)$;

**Output:** The gradient estimator $\frac{1}{2}[\hat{\nabla} f(x) + \tilde{\nabla} f(x)]$

---

**Proposition 4.1.** *Let $v$ be a random vector generated by Algorithm 1. Then it has the following properties: (a) $(v^\top a)^2 = \delta \|a\|^2$, and (b) $\mathbb{E}_{v \sim V} vv^\top = \delta I_d$.*

*Proof.* The proof is deferred to Appendix B.2. ∎

This proposition confirms that our sampling strategy yields the desired properties. These properties guarantee that the resulting output $v$ minimizes the variance of two-point gradient estimator. In the next section, we will empirically evaluate our practical implementations in both synthetic and real-world examples.

## 5  EXPERIMENTS

To validate our theoretical findings and demonstrate the practical effectiveness of DAPs in zeroth-order optimization, we conduct experiments on two different problem settings: synthetic examples

and language model optimization. For each target function $f$, we estimate its gradient using the zeroth-order estimator defined by different random perturbation under the batch size $b$:

$$\hat{\nabla} f(x) := \frac{1}{\mu b} \sum_{i=1}^{b} [f(x + \mu v_i) - f(x)] v_i, \tag{6}$$

where $\mu > 0$ is the perturbation size and $v_i$ are random vectors independently drawn from distributions according to the method used. For additional experiments, we refer the reader to Appendix E

### 5.1 SYNTHETIC EXAMPLES

We first evaluate our proposed method on the following two objective functions:

$$f_{\text{Quad}}(x) = x^\top A x, \quad f_{\text{Prod}}(x) = \prod_{i=1}^{d} x_i,$$

where each entry of $A \in \mathbb{R}^{d \times d}$ is independently sampled from the uniform distribution $U[0,1]$. The gradient of each objective function can be explicitly evaluated; that is, $\nabla f_{\text{Quad}}(x) = (A + A^\top)x$ and $\nabla f_{\text{Prod}}(x) = [x_2 x_3 \dots x_d, x_1 x_3 \dots x_d, \cdots, x_1 x_2 \dots x_{d-1}]$. We compare the performance of different random perturbations using the $\tau$-effective Mean-Square-Error ($\tau$-MSE), which is defined as

$$\tau\text{-MSE}(\hat{\nabla} f(x; v)) := [\hat{\nabla} f(x; v) - \nabla f(x)]^\top \Sigma_\tau [\hat{\nabla} f(x; v) - \nabla f(x)], \tag{7}$$

where $\Sigma_\tau$ is a diagonal matrix such that

$$\Sigma_\tau[i, i] = \begin{cases} 1 & |\nabla f(x)_i| > \tau \\ 0 & |\nabla f(x)_i| \leqslant \tau \end{cases}.$$

It represents the direction with a larger gradient value, which is of greater interest to us. In this experiment, we set $\tau = 10^{-4}$ for both figures in Figure 3. For the quadratic function, we force half of the entries in $x$ to be 0, while for the product function, we set the first element of $x$ to 0. This configuration ensures gradient sparsity; the product function represents an extremely sparse case where only one entry, $x_2 x_3 \dots x_d$, is non-zero. Our compared methods include the DAP with and without knowledge of the true gradient (Exact Gradient and Empirical Gradient), the Uniform Perturbation (where $v$ is uniformly distributed over the sphere with radius $\sqrt{d}$), and the Gaussian Perturbation (where $v \sim \mathcal{N}(0, I_d)$).

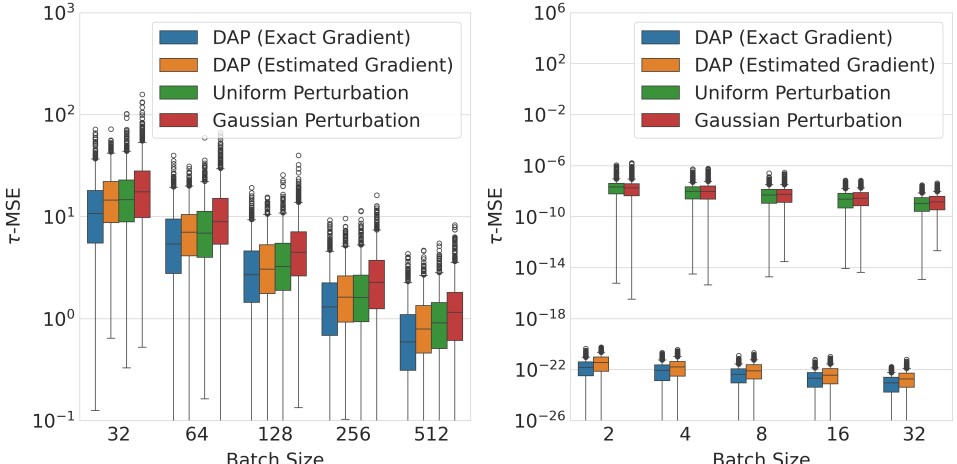

Figure 3: Comparison of $\tau$-MSE (defined in Eq. (7)) for different random perturbations on Quadratic (left) and Product (right) functions. Experiments were conducted with the dimension $d = 16$ and perturbation stepsize $\mu = 10^{-4}$. The x-axis shows the batch size, and the y-axis shows the $\tau$-MSE. Lower $\tau$-MSE indicates better gradient estimation accuracy along those more important directions.

The experiment yields three key insights: (1) When the gradient is known exactly, the DAP consistently outperforms classical random perturbations. As the batch size increases (to larger than $b = 32$ for the quadratic function), we observe the same phenomenon in the estimator generated using Algorithm 2. (2) When the gradient is extremely sparse, the DAP achieves significantly better accuracy along the effective direction.

## 5.2 LANGUAGE MODEL OPTIMIZATION

In this section, we demonstrate the practical applicability of the DAP in optimizing the neural network. We apply it to the task of fine-tuning a pre-trained language model. Using zeroth-order optimization to fine-tune the LLMs has been an active research field in recent years due to its effectiveness in saving memory (Malladi et al., 2023; Zhang et al., 2024; Gautam et al., 2024; Guo et al., 2024); it allows for the adjustment of model parameters without requiring access to the full computational graph, which can be prohibitively large for modern language models.

We conducted experiments using the OPT-1.3b model (Zhang et al., 2022) for sentiment classification on the Stanford Sentiment Treebank (SST-2) dataset (Socher et al., 2013). To ensure fair comparison, we maintained consistent parameters across experiments: learning rate $\eta = 10^{-4}$, perturbation size $\mu = 10^{-5}$, and batch size $b = 2$. Detailed experimental settings are provided in Appendix D. As shown in Figure 4, zeroth-order optimization using DAPs achieved superior performance compared to other random perturbation methods. Notably, we found that this superior performance does not rely on a large batch size $b$, demonstrating the practical effectiveness of DAP in real-world applications.

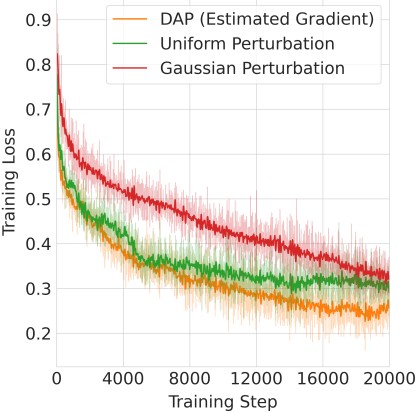

Figure 4: Performance comparison of different optimization methods for fine-tuning OPT-1.3b on SST-2. DAPs achieve empirically superior performance among other perturbations including the classical Gaussian smoothing and uniform smoothing.

## 6 CONCLUSION

In this work, we investigate zeroth-order optimization for smooth objective functions. We derive the conditions for random perturbations that minimize the variance of the two-point gradient estimator, revealing that in addition to traditional fixed-length random perturbations, directionally aligned perturbations (DAPs) can also achieve the minimum asymptotic variance as $\mu \to 0$. Our theoretical analysis extends best-known complexity bounds to encompass this broader class of perturbations, including DAPs. We explore the directionally aligned property of DAPs in gradient estimation and demonstrate their superior performance compared to traditional methods under specific conditions through experiments on both synthetic problems and language model fine-tuning tasks. These findings not only advance our understanding of zeroth-order optimization but also provide a new tools for improving gradient estimation. Our work opens up new avenues for research in zeroth-order methods and their applications in machine learning and optimization.

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

# Appendix

## Table of Contents

## A RELATED WORKS

### A.1 RELATED RESULTS IN ZEROTH-ORDER OPTIMIZATION

**Convergence Analysis for ZOO** The convergence of Stochastic Gradient Descent (SGD) has been extensively studied under various settings. Ghadimi & Lan (2013) established complexity results for computing approximate solutions using first-order and zeroth-order (gradient-free) information with Gaussian smoothing. For smooth convex objective functions, Duchi et al. (2015) obtained the optimal convergence upper bound for SGD under the zeroth-order optimization (ZOO) setting. Nesterov & Spokoiny (2017) provided the optimal convergence upper bound for Gaussian smoothing. In the realm of nonconvex optimization, Ji et al. (2019) proposed two new zeroth-order variance-reduced optimization algorithms, ZO-SVRG-Coord-Rand and ZO-SPIDER-Coord, and provided improved analysis for the existing ZO-SVRG-Coord algorithm. These methods achieved better convergence rates and function query complexities than previous approaches. Berahas et al. (2022) derived convergence analyses for finite differences, linear interpolation, Gaussian smoothing, and uniform sphere smoothing methods. Recent studies have focused on non-smooth settings. Davis et al. (2022) and Zhang et al. (2020) established the sample complexity for Lipschitz functions without assuming smoothness. Lin et al. (2022) derived the complexity upper bound of SGD while noting a $\sqrt{d}$ scale compared to the smooth setting. Notably, Rando et al. (2024a) and Kornowski & Shamir (2024) revealed that by applying certain techniques, the non-smooth case is not inherently more challenging than the smooth case.

**General Random Perturbations** Kozak et al. (2021) introduces a stochastic subspace descent algorithm for high-dimensional optimization problems with costly gradient evaluations. This method offers theoretical convergence guarantees and demonstrates a more general class of random perturbations that have convergence guarantees. Kozak et al. (2023); Rando et al. (2024b) later extends

this concept to a more generalized orthogonal subspace method, which also presents the optimal dependence on the dimension parameter $d$. In a related study, Duchi et al. (2015) examines random perturbations with finite fourth-order momentum, which aligns closely with our findings. However, their analysis is limited to convex optimization, whereas our work explores non-convex optimization scenarios.

## A.2 Common Choices of Random Perturbations

As observed in practice, the choice of distribution for the random perturbation vector $v$ potentially impacts the performance of zeroth-order gradient estimators. In this subsection, we review four classical distributions below:

**Gaussian Random Vector** The Gaussian (or normal) distribution is widely used due to its theoretical properties (Nesterov & Spokoiny, 2017) and ease of sampling. In this case, each component of $v$ is drawn independently from a standard normal distribution: $v_i \sim \mathcal{N}(0, 1)$ for $i = 1, \ldots, d$. More generally, the random vector $v$ can be sampled from a normal distribution with a given covariance matrix $\Sigma$; that is, $v \sim \mathcal{N}(0, \Sigma)$.

**Uniform Random Vector** Another common perturbation is the uniform random vector over the unit sphere. It has been shown in Duchi et al. (2015) that the uniform random vector achieves the optimal dependence on the dimension $d$. To sample such a random vector, it suffices to generate a vector with independent standard normal components and then normalize it to unit length. This method produces a vector uniformly distributed on the surface of the unit sphere.

**Rademacher Random Vector** The Rademacher distribution is widely used in the method known as Simultaneous Perturbation Stochastic Approximation (SPSA) (Spall, 1987; 1992). In this case, each component of $v$ is independently drawn from a Rademacher distribution, taking values $+1$ or $-1$ with equal probability. This distribution is particularly useful for its simplicity in certain optimization settings.

**Random Coordinate** The random coordinate method (Ji et al., 2019), also known as coordinate-wise perturbation, involves perturbing only one randomly selected coordinate at a time. This approach can be particularly effective in high-dimensional spaces where full-vector perturbations might be computationally expensive. To implement this, one randomly selects an index $i \in 1, \ldots, d$ and sets $v_i = 1$ and $v_j = 0$ for all $j \neq i$. This method can lead to more stable estimates in certain scenarios and is often used in large-scale optimization problems.

## B Derivation of the Minimum-Variance Random Perturbation

Our proof technique is mainly inspired by the following result taken from Theorem 1 (Móri et al., 1994), which indicates that $\mathbb{E}\|v\|^4$ is minimized when the distribution $V$ is the uniform distribution over the ball $\left\|v - \frac{s}{2}\right\|^2 = d + \left\|\frac{s}{2}\right\|^2$, where $s := \mathbb{E}_{v \sim V}(\|v\|^2 v)$. Here, we generalize its result by considering the $\delta$-unbiasedness structure $\mathbb{E}_{v \sim V} vv^\top = \delta I_d$.

**Lemma B.1.** *Let $V$ be any distribution over $\mathbb{R}^d$. Define $s := \mathbb{E}_{v \sim V}(\|v\|^2 v)$. Then*

$$\mathbb{E}\|v\|^4 \geqslant \delta^2 d^2 + \|s\|^2.$$

*Proof.* Let $f = \|v\|^2 - \delta d$ and $g = s^\top v$, where $v$ is a random vector from the distribution $V$ and $s := \mathbb{E}_{v \sim V}(\|v\|^2 v)$. Then

$$\|s\|^4 = (\mathbb{E}fg)^2 \leqslant \mathbb{E}f^2 \mathbb{E}g^2 = (\mathbb{E}\|v\|^4 - \delta^2 d^2) \|s\|^2 = (\mathbb{E}\mathrm{Tr}(vv^\top)^2 - \delta^2 d^2) \|s\|^2.$$

Then we obtain $\|s\|^2 + \delta^2 d^2 \leqslant \mathbb{E}\mathrm{Tr}(vv^\top)^2$. □

By constructing appropriate $f$ and $g$, we will obtain the desired estimation on $\mathbb{E}a^\top(vv^\top)^2 a$.

## B.1 PROOF OF THEOREM 2.2

*Proof.* The proof logic mainly follows Lemma B.1. Here, we construct the desired $f$ and $g$. First, we notice that the equality

$$(a^\top v)^2 = a^\top v v^\top a.$$

Let $f = (v^\top a)v - \delta a$ and $g = (\|v\|^2 - \delta d)a$. By these definitions, we have

$$\|f\|^2 = a^\top (v v^\top - \delta I_d)^2 a + \delta^2 \|a\|^2 - 2a^\top (v v^\top - \delta I_d)a$$

$$= a^\top (v v^\top)^2 a - 2\delta a^\top v v^\top a + \delta^2 \|a\|^2 - 2a^\top v v^\top a - 2\delta \|a\|^2$$

$$\mathbb{E}\|f\|^2 = a^\top \mathbb{E}(v v^\top - \delta I_d)^2 a - 2\delta^2 \|a\|^2 + \delta^2 \|a\|^2 - 2\delta \|a\|^2 - 2\delta \|a\|^2$$

$$= \mathbb{E}(v^\top a)^2 \|v\|^2 - \delta^2 \|a\|^2.$$

$$\|g\|^2 = \|a\|^2 (\|v\|^2 - \delta d)^2$$

$$\mathbb{E}\|g\|^2 = (\mathbb{E}\|v\|^4 - \delta^2 d^2)\|a\|^2.$$

$$f^\top g = a^\top (v v^\top - \delta I_d)a \cdot (\|v\|^2 - \delta d)$$

$$= [(a^\top v)^2 - \delta \|a\|^2] \cdot (\|v\|^2 - \delta d)$$

$$= (a^\top v)^2 |v\|^2 - \delta \|a\|^2 \|v\|^2 - \delta d(a^\top v)^2 + \delta d\|a\|^2$$

$$\mathbb{E}f^\top g = \mathbb{E}[(a^\top v)^2 |v\|^2] - d\delta^2 \|a\|^2 - d\delta^2 \|a\|^2 + d\delta^2 \|a\|^2$$

$$= \mathbb{E}(a^\top v)^2 |v\|^2 - d\delta^2 \|a\|^2.$$

Then we obtain

$$[\mathbb{E}f^\top g]^2 = \left[\mathbb{E}(v^\top a)^2 \|v\|^2 - \delta^2 d\|a\|^2\right]^2$$

$$\overset{(i)}{\leqslant} \mathbb{E}\|f\|^2 \mathbb{E}\|g\|^2$$

$$= \left[\mathbb{E}(v^\top a)^2 \|v\|^2 - \delta^2 \|a\|^2\right] \left[(\mathbb{E}\|v\|^4 - \delta^2 d^2)\|a\|^2\right]$$

where (i) applies the Cauchy-Schwarz inequality. For convenience, we define

$$X := \mathbb{E}(v^\top a)^2 \|v\|^2.$$

Then this inequality is simplified as:

$$[X - \delta^2 d\|a\|^2]^2 \leqslant [X - \delta^2 \|a\|^2][(\mathbb{E}\|v\|^4 - \delta^2 d^2)\|a\|^2].$$

$$\iff X^2 + \delta^4 d^2 \|a\|^4 - 2\delta^2 d\|a\|^2 X \leqslant \left[(\mathbb{E}\|v\|^4 - \delta^2 d^2)\|a\|^2\right] X - \delta^2 \|a\|^2 \left[(\mathbb{E}\|v\|^4 - \delta^2 d^2)\|a\|^2\right].$$

$$\iff X^2 - 2\delta^2 d\|a\|^2 X \leqslant \left[(\mathbb{E}\|v\|^4 - \delta^2 d^2)\|a\|^2\right] X - \delta^2 \|a\|^4 \mathbb{E}\|v\|^4.$$

$$\iff X^2 - \left[2\delta^2 d\|a\|^2 + \left[(\mathbb{E}\|v\|^4 - \delta^2 d^2)\|a\|^2\right]\right] X \leqslant -\delta^2 \|a\|^4 \mathbb{E}\|v\|^4.$$

Let $2C = 2\delta^2 d\|a\|^2 + \left[(\mathbb{E}\|v\|^4 - \delta^2 d^2)\|a\|^2\right]$. Then

$$C^2 - \delta^2 \|a\|^4 \mathbb{E}\|v\|^4 = \delta^4 d^2 \|a\|^4 + \frac{\|a\|^4}{4} \left[\mathbb{E}\|v\|^4 - \delta^2 d^2\right]^2 + \delta^2 d\|a\|^4 \left[\mathbb{E}\|v\|^4 - \delta^2 d^2\right]$$

$$- \delta^2 \|a\|^4 \left[\mathbb{E}\|v\|^4 - \delta^2 d^2 + \delta^2 d^2\right]$$

$$= \frac{\|a\|^4}{4} \left[\mathbb{E}\|v\|^4 - \delta^2 d^2\right]^2 + \delta^2 d\|a\|^4 \left[\mathbb{E}\|v\|^4 - \delta^2 d^2\right] - \delta^2 \|a\|^4 \left[\mathbb{E}\|v\|^4 - \delta^2 d^2\right]$$

$$= \frac{\|a\|^4}{4} \left[\mathbb{E}\|v\|^4 - \delta^2 d^2\right]^2 + \delta^2 (d-1)\|a\|^4 \left[\mathbb{E}\|v\|^4 - \delta^2 d^2\right] \geqslant 0.$$

Therefore, we obtain an upper bound and a lower bound for $X$:

$$X \leqslant C + \sqrt{C^2 - \delta^2 \|a\|^4 \mathbb{E}\|v\|^4},$$

$$X \geqslant C - \sqrt{C^2 - \delta^2 \|a\|^4 \mathbb{E}\|v\|^4} \overset{(i)}{\geqslant} \delta^2 d\|a\|^2.$$

Here, (i) additionally applies the following statement: We assume $X = \delta^2 d\|a\|^2 - c < \delta^2 d\|a\|^2$ for some distribution $V$ and an error term $c > 0$. Then we conclude that $\mathbb{E}\|v\|^4 < \delta^2 d^2$, which is impossible. The detailed proof is given as follows:

Since $\mathbb{E}a^\top(vv^\top vv^\top)a = \delta^2 d\|a\|^2 - c$, for an orthogonal linear transformation $O$, we have

$$\mathbb{E}(O^k a)^\top\Big(vv^\top vv^\top - (\delta^2 d\|a\|^2 - c)I_d\Big)O^k a = 0,$$

for $k = 1, 2, \ldots, d - 1$. Because $\{O^k a\}_{k=0}^d$ forms a basis of $\mathbb{R}^d$, we conclude $\mathbb{E}\Big(vv^\top vv^\top - (\delta^2 d\|a\|^2 - c)I_d\Big) = 0_d$ by using Lemma B.2, where $0_d$ represents the zero matrix with the dimension $d \times d$. This further leads to

$$\mathrm{Tr}\mathbb{E}\Big(vv^\top vv^\top - (\delta^2 d\|a\|^2 - c)I_d\Big) = 0_d$$
$$\implies \mathbb{E}\|v\|^4 = \delta^2 d^2\|a\|^2 - cd < \delta^2 d^2\|a\|^2.$$

Therefore, $X < \delta^2 d\|a\|^2$ leads to a contradiction.

In the remaining of this proof, we will derive the equality condition. By the Cauchy–Schwarz inequality, the equality holds *if and only if* $f = 0$, $g = 0$, or $f = rg$ for some $r > 0$. We discuss each of condition separately. Throughout this discussion, we assume $a \neq 0$.

- $f = 0$: That is, $(v^\top a)v - \delta a = 0$ holds almost surely. By timing $a^\top$ on both sides, this condition becomes
$$(a^\top v)^2 = \delta\|a\|^2.$$

- $g = 0$: That is, $(\|v\|^2 - \delta d)a = 0$ holds almost surely. By setting $A = (\|v\|^2 - \delta d)I_d$ in Lemma B.6, this equality holds if and only if
$$\|v\|^2 = \delta d.$$

- $f = rg$ for some $r > 0$: That is,
$$(v^\top a)v - \delta a = r(\|v\|^2 - \delta d)a.$$
We will show that this condition cannot hold. Assume it holds for some $r$, then for the case $v^\top a \neq 0$, we have
$$(v^\top a)^2 - \delta a^\top v = r(\|v\|^2 - \delta d)a^\top v$$
$$\implies v^\top a - \delta = r(\|v\|^2 - \delta d)$$
$$\implies \mathbb{E}v^\top a - \delta = r(\mathbb{E}\|v\|^2 - \delta d)$$
$$\implies -\delta = 0.$$
For the case $v^\top a = 0$, we directly take the expectation on both sides:
$$-\delta a = 0.$$

Then the proof is completed. $\qquad\square$

## B.2 PROOF OF PROPOSITION 4.1

*Proof.* We prove parts (a) and (b) of the proposition separately.

For part (a), we show that $(v^\top a)^2 = \delta\|a\|^2$. After projecting $v_\mathrm{ini}$ onto the plane
$$\mathcal{P}_\xi = v \in \mathbb{R}^d : a^\top v = \xi\sqrt{\delta}\|a\|,$$
the resulting vector $v$ satisfies $a^\top v = \xi\sqrt{\delta}\|a\|$. Squaring both sides yields $(v^\top a)^2 = (a^\top v)^2 = (\xi\sqrt{\delta}\|a\|)^2 = \delta\|a\|^2$, proving part (a).

For part (b), we prove that $\mathbb{E}_{v \sim V}[vv^\top] = \delta I_d$. We express $v$ in terms of $v_\mathrm{ini}$ as $v = v_\mathrm{ini} - (a^\top v_\mathrm{ini} - \xi\sqrt{\delta}\|a\|)\frac{a}{\|a\|^2}$. Let $u = \frac{a}{\|a\|}$ be the unit vector in the direction of $a$. We can rewrite $v$ as $v = w + \xi\sqrt{\delta}u$, where $w = v_\mathrm{ini} - (u^\top v_\mathrm{ini})u$ is orthogonal to $u$. Computing $vv^\top$ and taking the expectation, we get $\mathbb{E}[vv^\top] = \mathbb{E}[ww^\top] + \delta uu^\top$. The cross terms vanish due to the independence of $\xi$ and $v_\mathrm{ini}$, and because $\mathbb{E}[\xi] = 0$. Expanding $ww^\top$ and taking expectations, using $\mathbb{E}[v_\mathrm{ini}] = 0$ and $\mathbb{E}[v_\mathrm{ini}v_\mathrm{ini}^\top] = \delta I_d$, we obtain:
$$\mathbb{E}[ww^\top] = \delta I_d - \delta uu^\top - \delta uu^\top + \delta uu^\top = \delta I_d - \delta uu^\top.$$

Finally, we have $\mathbb{E}[vv^\top] = \mathbb{E}[ww^\top] + \delta uu^\top = \delta I_d - \delta uu^\top + \delta uu^\top = \delta I_d$, completing the proof of part (b) and the proposition. $\qquad\square$

### B.3 Supporting Lemmas

The following result can be found in Hungerford (2012). We omit its proof here.

**Lemma B.2.** *Let $A \in \mathbb{R}^{d \times d}$ be a matrix. Suppose $x^\top A x = 0$ for all $x \in \mathbb{R}^d$, then $A$ must be a skew-symmetric matrix (i.e. $A^\top = -A$). Moreover, if $A$ is a symmetric matrix (i.e. $A^\top = A$, then $A$ must be a zero matrix.*

A stronger version of the following lemma is given by Mirsky (1975). The equality condition can be found in Rhea (2011).

**Lemma B.3.** *Let $A$ and $B$ be positive semidefinite. Then*
$$\mathrm{Tr}(AB) \leqslant \mathrm{Tr}(A)\mathrm{Tr}(B).$$

*Proof.* Let $\alpha = \mathrm{Tr}(A)$. Then
$$\mathrm{Tr}(AB) \overset{(i)}{=} \mathrm{Tr}(B^{1/2}AB^{1/2}) \overset{(ii)}{\leqslant} \mathrm{Tr}(B^{1/2}(\alpha I)B^{1/2}) = \mathrm{Tr}(A)\mathrm{Tr}(B).$$
Here, (i) applies the semidefinite of $B$ and $\mathrm{Tr}(AB) = \mathrm{Tr}(BA)$; (ii) applies the operator monotonicity of the trace operator. $\qquad\square$

The following lemma gives the explicit representation of the projection of a vector $v \in \mathbb{R}^d$ on a hyper-plane in $\mathbb{R}^d$. This result can also be found in Seber & Lee (2012).

**Lemma B.4.** *Let $\mathcal{P} := \{v : u^\top v = c\} \subset \mathbb{R}^d$ be a hyper-plane associated with fixed vector $u \in \mathbb{R}^d$ and a constant $c \in \mathbb{R}$. For a given vector $v \in \mathbb{R}^d$, suppose that $\mathrm{Proj}(v)$ denotes the orthogonal projection of $v$ onto $\mathcal{P}$. Then*
$$\mathrm{Proj}(v) := v - u(u^\top v - c)/\|u\|_2^2.$$

*Proof.* The projection is given by the following optimization problem
$$\min_{w \in \mathbb{R}^d} \quad \frac{1}{2}\|w - v\|^2$$
$$\text{s.t.} \quad w^\top u = c.$$
The Lagrangian is given by
$$\mathcal{L}(w, \lambda) = \frac{1}{2}\|w - v\|^2 + \lambda(w^\top u - c).$$
The projection is solved as
$$\mathrm{Proj}(v) = v - u(v^\top u - c)/\|u\|^2.$$
$\qquad\square$

**Lemma B.5.** *Let $v$ be a random vector following the distribution $V$ and all entries are mutually independent. Then*
$$\mathbb{E}_{v \sim V}\big(\mathrm{Tr}((vv^\top)^2)\big) = \sum_{i=1}^d \mathbb{E}[v_i^4] + d(d-1).$$

*Proof.* To prove this result, it requires to explicitly evaluate $\mathbb{E}_{v \sim V}\big(\mathrm{Tr}((vv^\top)^2)\big)$. Let
$$v = [v_1, v_2, \ldots, v_d]^\top.$$
We will evaluate $\mathbb{E}\big((vv^\top)^2\big)$ by considering the $i$-th row and $j$-th column of $vv^\top$:
$$(vv^\top)i\cdot = \big[v_i v_1, v_i v_2, \ldots, v_i^2, \ldots, v_i v_d\big],$$
$$(vv^\top)\cdot j = \begin{bmatrix} v_1 v_j \\ v_2 v_j \\ \vdots \\ v_j^2 \\ \vdots \\ v_d v_j \end{bmatrix}.$$

Then, the $(i, j)$-th entry of $(vv^\top)^2$ is

$$(vv^\top)^2_{i,j} = \begin{bmatrix} v_i v_1, v_i v_2, \ldots, v_i^2, \ldots, v_i v_d \end{bmatrix} \begin{bmatrix} v_1 v_j \\ v_2 v_j \\ \vdots \\ v_j^2 \\ \vdots \\ v_d v_j \end{bmatrix} = \sum_{k=1}^d v_i v_j v_k^2.$$

Taking the expectation on both sides and applying the independence, we obtain

$$\mathbb{E}\big[(vv^\top)^2_{i,j}\big] = \begin{cases} 0 & \text{if } i \neq j, \\ \mathbb{E}[v_i^4] + (d-1) & \text{if } i = j. \end{cases}$$

Here, the constraint $\mathbb{E}[vv^\top] = I_d$ implies that $\mathbb{E}[v_i^2] = 1$ for all $i$. Therefore, we have

$$\begin{aligned}
\mathbb{E}\big(\mathrm{Tr}\big((vv^\top)^2\big)\big) &= \sum_{i=1}^d \mathbb{E}\big[(vv^\top)^2_{i,i}\big] \\
&= \sum_{i=1}^d \big(\mathbb{E}[v_i^4] + (d-1)\big) \\
&= \sum_{i=1}^d \mathbb{E}[v_i^4] + d(d-1).
\end{aligned}$$

$\square$

**Lemma B.6.** *Let $a \in \mathbb{R}^d$ and $A$ be a positive semi-definite matrix. Then $a^\top A a = 0$ if and only if $Aa = 0$.*

*Proof.* Since $A$ is positive semi-definite, we can represent $A$ as $A = B^2$, where $B$ is also a positive semi-definite matrix. Then

$$a^\top A a = (a^\top B)^2 = \|a^\top B\|^2 = 0.$$

By the definition of vector norm, this equality holds if and only if $a^\top B = 0$. We multiply $B$ on both sides and obtain $Aa = 0$. $\square$

## C    Convergence Analysis of Zeroth-Order SGD

Our proof is mainly adapted from Khaled & Richtárik (2022) and Mishchenko et al. (2020) with additionally considering the variance introduced by gradient estimation.

**Assumption C.1.** *In the optimization problem given by Eq. (1), the individual loss function $f(\cdot; \xi) : \mathbb{R}^d \to \mathbb{R}$ satisfies the following two properties:*

*(a) L-Smoothness; for all $x, y \in \mathbb{R}^d$,*

$$f(y; \xi) \leqslant f(x; \xi) + \nabla f(x; \xi)^T (y - x) + \frac{L}{2} \|y - x\|^2.$$

*(b) Lower boundedness; the infimum $f_\xi^* := \inf_{x \in \mathbb{R}^d} f(x; \xi)$ exists almost surely with $\xi \sim \Xi$.*

Though this assumption could be further weakened to the expected smoothness assumption on the objective function (Khaled & Richtárik, 2022), the current version has covered a sufficiently broad class of functions, including many neural network structures. By Rademacher's theorem (Evans & Gariepy, 2015), the $L$-smoothness implies the almost-everywhere differentiability of the gradient $\nabla f(x; \xi)$, which makes the standard Taylor theorem directly available for the individual loss $f(x; \xi)$.

For strongly convex settings, a faster convergence rate is often guaranteed; it relies on an additional assumption on the objective function $f(x) := \mathbb{E}_{\xi \sim \Xi} f(x; \xi)$:

**Assumption C.2.** *In the optimization problem given by Eq. (1), the objective loss function $f(x) := \mathbb{E}_{\xi \sim \Xi} f(x; \xi)$ satisfies the $c$-strongly convexity property; that is, for all $x, y \in \mathbb{R}^d$,*

$$f(y) \geq f(x) + \nabla f(x)^T (y - x) + \frac{c}{2} \|y - x\|^2.$$

We note that this assumption could be further relaxed to the Polyak-Łojasiewicz condition $f(x) - f^* \leq c_{\text{PL}} \|\nabla f(x)\|^2$ (Karimi et al., 2016).

## C.1   PROOF OF THEOREM 3.1

In this subsection, we present the main proof for Theorem 3.1. Theorem C.3 gives the proof for Part (a) and Theorem C.4 gives the proof for Part (b). First, we re-state these theorem as follows:

**Theorem C.3.** *Suppose that Assumption C.1 and Assumption 2.1 are satisfied. Let $\{x_t\}$ be the SGD dynamic solving Eq. (1) generated by the update rule Eq. (4). If the learning rate $\eta \leq \min\{\frac{1}{2L}, \frac{1}{L\sqrt{2T(2\delta^2 d + \rho_V + 2\delta + 1)}}\}$, then*

$$\min_{1 \leq t \leq T} \mathbb{E}\|\nabla f(x_t)\|^2 \leq \frac{(f(x_1) - f^*)}{\delta \eta T} + \frac{2\eta}{\delta} \Big[ LB^2(1 + \beta_V) + \mu^2 \alpha_V \Big],$$

*where $\rho_V := \mathbb{E}\|v\|^4 - \delta^2 d^2$, $\alpha_V := L^3 \mathbb{E}\|v\|^4$, $\beta_V := 2\delta^2 d + \rho_V + 1 - 2\delta$, and $B^2 := 2L(f^* - \mathbb{E}_{\xi \sim \Xi} f_\xi^*)$.*

*Proof.* We start from Eq. (11) from Lemma C.8:

$$\frac{\delta \eta}{2} \mathbb{E}\|\nabla f(x_t)\|^2 \leq \big(1 + 4L^2 \eta^2 + 2L^2 \beta_V \eta^2\big)\big(\mathbb{E}f(x_t) - f^*\big) - \big(\mathbb{E}f(x_{t+1}) - f^*\big)$$
$$+ \eta^2 LB^2(1 + \beta_V) + \eta^2 \mu^2 \alpha_V,$$

For convenience, we define $\delta_t = \mathbb{E}f(x_t) - f^*$, $M_t = \eta^2 LB^2(1 + \beta_V) + \eta^2 \mu^2 \alpha_V$, and $c = 1 + 4L^2 \eta^2 + 2L^2 \beta_V \eta^2$. Then

$$\frac{\delta \eta}{2} \mathbb{E}\|\nabla f(x_T)\|^2 \leq c \times \delta_T - \delta_{T+1} + M_T$$

$$c \times \frac{\delta \eta}{2} \mathbb{E}\|\nabla f(x_{T-1})\|^2 \leq c^2 \times \delta_{T-1} - c \times \delta_T + c \times M_{T-1}$$

$$\vdots$$

$$c^{T-1} \times \frac{\delta \eta}{2} \mathbb{E}\|\nabla f(x_1)\|^2 \leq c^T \times \delta_1 - c^T \times \delta_1 + c^{T-1} \times M_1.$$

We sum all together. Then

$$\Big(\sum_{i=0}^{T-1} c^i\Big) \frac{\delta \eta}{2} \min_{1 \leq t \leq T} \mathbb{E}\|\nabla f(x_t)\|^2 \leq c^T \times \delta_0 + \Big(\sum_{i=0}^{T-1} c^i\Big) \max_{1 \leq t \leq T} M_t.$$

Putting back the shortcut notations ($\delta_t = \mathbb{E}f(x_t) - f^*$, $M_t = \eta^2 LB^2(1 + \beta_V) + \eta^2 \mu^2 \alpha_V$, and $c = 1 + 4L^2 \eta^2 + 2L^2 \beta_V \eta^2$), the above inequality solves the upper bound of $\min_t \mathbb{E}\|\nabla f(x_t)\|^2$ as

$$\min_{1 \leq t \leq T} \mathbb{E}\|\nabla f(x_t)\|^2 \leq \frac{2}{\delta \eta} \frac{c^T}{\sum_{i=0}^{T-1} c^i}\big(f(x_1) - f^*\big) + \frac{2}{\delta \eta} \max_{1 \leq t \leq T} M_t$$

$$= \frac{2}{\delta \eta} \frac{\big(1 + 4L^2 \eta^2 + 2L^2 \beta_V \eta^2\big)^T}{\sum_{i=0}^{T-1} \big(1 + 4L^2 \eta^2 + 2L^2 \beta_V \eta^2\big)^i}\big(f(x_1) - f^*\big) + \frac{2}{\delta \eta}\Big[\eta^2 LB^2(1 + \beta_V) + \eta^2 \mu^2 \alpha_V\Big]$$

$$= \frac{2\big(4L^2 \eta^2 + 2L^2 \beta_V \eta^2\big)}{\delta \eta} \frac{\big(1 + 4L^2 \eta^2 + 2L^2 \beta_V \eta^2\big)^T}{\big(1 + 4L^2 \eta^2 + 2L^2 \beta_V \eta^2\big)^T - 1}\big(f(x_1) - f^*\big)$$

$$+ \frac{2\eta}{\delta}\Big[LB^2(1+\beta_V) + \mu^2\alpha_V\Big]$$

$$\overset{(i)}{\leqslant} \frac{4L^2\eta\big(2+\beta_V\big)}{\delta} \frac{e^{T\big(4L^2\eta^2 + 2L^2\beta_V\eta^2\big)}}{T\big(4L^2\eta^2 + 2L^2\beta_V\eta^2\big)}\big(f(x_1) - f^*\big) + \frac{2\eta}{\delta}\Big[LB^2(1+\beta_V) + \mu^2\alpha_V\Big]$$

$$\overset{(ii)}{\leqslant} \frac{\big(f(x_1) - f^*\big)}{\delta\eta T} + \frac{2\eta}{\delta}\Big[LB^2(1+\beta_V) + \mu^2\alpha_V\Big],$$

where (i) applies $1 + x \leqslant e^x$ and $(1+x)^T - 1 \geqslant Tx$, (ii) applies the condition

$$\eta \leqslant \frac{1}{\sqrt{T\big(4L^2 + 2L^2\beta_V\big)}}$$

on the learning rate $\eta$, $B$ is given in Lemma C.6, and $\alpha_V$, $\beta_V$ are given in Lemma C.8. $\qquad\square$

**Theorem C.4.** *Suppose that Assumption C.1, Assumption C.2, and Assumption 2.1 are satisfied. Let $\{x_t\}$ be the SGD dynamic solving Eq. (1) generated by the update rule Eq. (4). If the learning rate $\eta \leqslant \min\{\frac{1}{2L}, \frac{\delta c}{4L^2} \frac{1}{2\delta^2 d + 2\delta + 1 + \rho_V}\}$, then*

$$\mathbb{E}f(x_T) - f^* \leqslant \big(1 - \frac{c}{2}\delta\eta\big)^{T-1}\big(f(x_1) - f^*\big) + \frac{2}{c\delta}\eta\big(LB^2(1+\beta_V) + \mu^2\alpha_V\big),$$

*where $\rho_V := \mathbb{E}\|v\|^4 - \delta^2 d^2$, $\alpha_V := L^3\mathbb{E}\|v\|^4$, $\beta_V := 2\delta^2 d + \rho_V + 1 - 2\delta$, and $B^2 := 2L\big(f^* - \mathbb{E}_{\xi \sim \Xi} f_\xi^*\big)$.*

*Proof.* We additionally apply the $c$-strong convexity assumption made on the objective function to the left-hand-side of Lemma C.8:

$$\delta\eta c\big(\mathbb{E}f(x_t) - f^*\big) \leqslant \big(1 + 4L^2\eta^2 + 2L^2\beta_V\eta^2\big)\big(\mathbb{E}f(x_t) - f^*\big) - \big(\mathbb{E}f(x_{t+1}) - f^*\big)$$
$$+ \eta^2(LB^2 + \beta_V) + \eta^2\mu^2\alpha_V$$

For convenience, we define $\delta_t = \mathbb{E}f(x_t) - f^*$, $M_t = \eta^2 LB^2(1+\beta_V) + \eta^2\mu^2\alpha_V$, and $r = 1 - c\delta\eta + \eta^2\big(4L^2 + 2L^2\beta_V\big)$. Then re-arranging this inequality leads to

$$\delta_{t+1} \leqslant r\delta_t + M_t$$
$$\vdots$$
$$\leqslant r^t\delta_1 + \sum_{i=0}^{t} r^i M_{t-i}.$$

Putting back the shortcut notations, the above inequality solves the upper bound of $\mathbb{E}f(x_T) - f^*$ as

$$\mathbb{E}f(x_T) - f^* \leqslant r^{T-1}\big(f(x_1) - f^*\big) + \frac{1}{1-r}M$$
$$\overset{(i)}{\leqslant} \big(1 - \frac{c}{2}\delta\eta\big)^{T-1}\big(f(x_1) - f^*\big) + \frac{\eta^2 LB^2(1+\beta_V) + \eta^2\mu^2\alpha_V}{c\delta\eta - \eta^2\big(4L^2 + 2L^2\beta_V\big)}$$
$$\leqslant \big(1 - \frac{c}{2}\delta\eta\big)^{T-1}\big(f(x_1) - f^*\big) + \frac{2}{c\delta}\eta\big(LB^2(1+\beta_V) + \mu^2\alpha_V\big),$$

where (i) applies the condition $\eta \leqslant \frac{\delta c}{4L^2} \frac{1}{2\delta^2 d + 2\delta + 1 + \rho_V}$. $\qquad\square$

## C.2 SUPPORTING LEMMAS

The following two lemmas (Lemma C.5 and Lemma C.6) are directly taken from Proposition 2, Mishchenko et al. (2020). We adapt the statement to our notations and give an explicit expression for $A$ and $B$ in Lemma C.6.

**Lemma C.5.** *Suppose that [Assumption C.1](#) is satisfied. Then the objective function $f(x) := \mathbb{E}_{\xi \sim \Xi} f(x; \xi)$ is also L-smooth and lower bounded; that is,*

*(a) for all $x, y \in \mathbb{R}^d$,*

$$f(y) \le f(x) + \nabla f(x)^T (y - x) + \frac{L}{2} \|y - x\|^2.$$

*(b) The infimum $f^* := \inf_{x \in \mathbb{R}^d} f(x)$ exists almost surely.*

**Lemma C.6.** *Suppose that [Assumption C.1](#) is satisfied. Then for any $x \in \mathbb{R}^d$,*

$$\mathbb{E}\|\nabla f(x; \xi) - \nabla f(x)\|^2 \le 2A(f(x) - f^*) + B^2, \tag{8}$$

*where the constants $A = 2L$ and $B = \sqrt{2L(f^* - \mathbb{E}_{\xi \sim \Xi} f_\xi^*)}$.*

*Proof.* By Lemma 1 from [Khaled & Richtárik (2022)](#), [Assumption C.1](#) (a) implies that

$$\|\nabla f(x; \xi)\|^2 \le 2L(f(x; \xi) - f_\xi^*);$$
$$\|\nabla f(x)\|^2 \le 2L(f(x) - f^*).$$

Summing them together leads to

$$\mathbb{E}\|\nabla f(x; \xi) - \nabla f(x)\|^2 = \mathbb{E}\|\nabla f(x; \xi)\|^2 + \mathbb{E}\|\nabla f(x)\|^2$$
$$\le 4L(f(x) - f^*) + 2L(f^* - \mathbb{E}_{\xi \sim \Xi} f_\xi^*).$$

Defining $A := 2L$ and $B = \sqrt{2L(f^* - \mathbb{E}_{\xi \sim \Xi} f_\xi^*)}$ concludes the proof. $\qquad\square$

The following lemma ([Lemma C.7](#)) builds the per-iteration recursion.

**Lemma C.7.** *Suppose that [Assumption C.1](#) is satisfied. Let $\{x_t\}$ be the SGD dynamic solving [Eq. (1)](#) generated by the update rule [Eq. (4)](#). If the learning rate $\eta \le \frac{1}{2L}$, then*

$$\frac{\delta \eta}{2} \mathbb{E}\|\nabla f(x_t)\|^2 \le (1 + 2AL\eta^2)(\mathbb{E}f(x_t) - f^*) - (\mathbb{E}f(x_{t+1}) - f^*) \tag{9}$$
$$+ \eta^2 L B^2 + \eta^2 L \mathbb{E} \mathsf{L}_t,$$

*where A and B are given in [Lemma C.6](#), and*

$$\mathsf{L}_t := \|\hat{\nabla} f(x_t; \xi_t, v_t) - \nabla f(x_t; \xi_t)\|^2 \tag{10}$$

*denote the accuracy of $\hat{\nabla} f(x_t; \xi_t, v_t)$ for estimating $\nabla f(x_t; \xi_t)$.*

*Proof.* Starting with $L$-smoothness of the objective function $f$ ([Lemma C.5](#)), we obtain

$$f(x_{t+1}) \le f(x_t) + \langle \nabla f(x_t), x_{t+1} - x_t \rangle + \frac{L}{2} \|x_{t+1} - x_t\|^2$$

$$= f(x_t) - \eta \langle \nabla f(x_t), \hat{\nabla} f(x_t; \xi_t, v_t) \rangle + \frac{L\eta^2}{2} \|\hat{\nabla} f(x_t; \xi_t, v_t)\|^2$$

$$\overset{(i)}{=} f(x_t) - \frac{\eta}{2} \left[ \|\nabla f(x_t)\|^2 + \|\hat{\nabla} f(x_t; \xi_t, v_t)\|^2 - \|\nabla f(x_t) - \hat{\nabla} f(x_t; \xi_t, v_t)\|^2 \right]$$

$$+ \frac{L\eta^2}{2} \|\hat{\nabla} f(x_t; \xi_t, v_t)\|^2$$

$$= f(x_t) - \frac{\eta}{2} \|\nabla f(x_t)\|^2 - \frac{\eta}{2}(1 - L\eta) \|\hat{\nabla} f(x_t; \xi_t, v_t)\|^2 + \frac{\eta}{2} \|\nabla f(x_t) - \hat{\nabla} f(x_t; \xi_t, v_t)\|^2$$

$$= f(x_t) - \frac{\eta}{2} \|\nabla f(x_t)\|^2 - \frac{\eta}{2}(1 - L\eta) \|\hat{\nabla} f(x_t; \xi_t, v_t) - \nabla f(x_t) + \nabla f(x_t)\|^2$$

$$+ \frac{\eta}{2} \|\nabla f(x_t) - \hat{\nabla} f(x_t; \xi_t, v_t)\|^2$$

$$\overset{(ii)}{=} f(x_t) - \frac{\eta}{2}\|\nabla f(x_t)\|^2 - \frac{\eta}{2}(1 - L\eta)\left[\|\hat{\nabla} f(x_t; \xi_t, v_t) - \nabla f(x_t)\|^2 + \|\nabla f(x_t)\|^2 \right.$$

$$\left. + 2\langle \hat{\nabla} f(x_t; \xi_t) - \nabla f(x_t), \nabla f(x_t)\rangle\right] + \frac{\eta}{2}\|\nabla f(x_t) - \hat{\nabla} f(x_t; \xi_t, v_t)\|^2$$

$$\mathbb{E}_{\xi_t} f(x_{t+1}) \overset{(iii)}{\leqslant} \mathbb{E}_{\xi_t} f(x_t) - \left(\eta - \frac{\eta^2 L}{2}\right)\mathbb{E}_{\xi_t}\|\nabla f(x_t)\|^2 - \eta(1 - L\eta)\mathbb{E}_{\xi_t}\langle \hat{\nabla} f(x_t; v) - \nabla f(x_t), \nabla f(x_t)\rangle$$

$$+ \frac{\eta^2 L}{2}\mathbb{E}_{\xi_t}\|\nabla f(x_t) - \hat{\nabla} f(x_t; \xi_t, v_t)\|^2$$

$$\overset{(iv)}{\leqslant} \mathbb{E}_{\xi_t} f(x_t) - \left(\eta - \frac{\eta^2 L}{2}\right)\mathbb{E}_{\xi_t}\|\nabla f(x_t)\|^2 - \eta(1 - L\eta)\mathbb{E}_{\xi_t}\nabla f(x_t)^\top\left(v_t v_t^\top - I\right)\nabla f(x_t)$$

$$+ \mu\frac{L}{2}\|v_t\|^2 v_t^\top \nabla f(x_t) + \frac{\eta^2 L}{2}\mathbb{E}_{\xi_t}\|\nabla f(x_t) - \hat{\nabla} f(x_t; \xi_t, v_t)\|^2$$

$$\mathbb{E} f(x_{t+1}) \leqslant \mathbb{E} f(x_t) - \left(\eta - \frac{\eta^2 L}{2}\right)\mathbb{E}\|\nabla f(x_t)\|^2 - \eta(1 - L\eta)(\delta - 1)\mathbb{E}\|\nabla f(x_t)\|^2$$

$$+ \frac{\eta^2 L}{2}\mathbb{E}\|\nabla f(x_t) - \hat{\nabla} f(x_t; \xi_t, v_t)\|^2$$

$$\overset{(v)}{\leqslant} \mathbb{E} f(x_t) - \left(\frac{\delta\eta}{2}\right)\mathbb{E}\|\nabla f(x_t)\|^2 + \frac{\eta^2 L}{2}\mathbb{E}\|\nabla f(x_t) - \hat{\nabla} f(x_t; \xi_t, v_t)\|^2,$$

where (i) applies the identity $2\langle a, b\rangle = \|a\|^2 + \|b\|^2 - \|a - b\|^2$, (ii) applies the identity $\|a + b\|^2 = \|a\|^2 + \|b\|^2 + 2\langle a, b\rangle$, (iii) takes the expectation with respect to the data distribution $\xi_t \sim \Xi$, (iv) applies the Taylor theorem (Lemma C.9) to expand $f(x_t + \mu v_t) - f(x_t)$ around $x_t$, and (v) uses $\delta\eta + \frac{\eta^2 L}{2} - \delta L\eta^2 \geqslant \frac{\delta\eta}{2}$ when $\eta \leqslant \frac{1}{2L}$. We further notice that

$$\mathbb{E}\|\nabla f(x_t) - \hat{\nabla} f(x_t; \xi_t, v_t)\|^2 = \mathbb{E}\|\nabla f(x_t) - \nabla f(x_t; \xi_t) + \nabla f(x_t; \xi_t) - \hat{\nabla} f(x_t; \xi_t, v_t)\|^2$$

$$\overset{(i)}{\leqslant} 2\mathbb{E}\|\nabla f(x_t) - \nabla f(x_t; \xi_t)\|^2 + 2\mathbb{E}\|\nabla f(x_t; \xi_t) - \hat{\nabla} f(x_t; \xi_t, v_t)\|^2$$

$$\overset{(ii)}{\leqslant} 4A\mathbb{E}(f(x_t) - f^*) + 2B^2 + 2\mathbb{E}\mathsf{L}_t,$$

where (i) applies $\|a + b\|^2 \leqslant 2\|a\|^2 + 2\|b\|^2$ and (ii) applies Lemma C.6. Rearranging the inequality, we obtain

$$\frac{\delta\eta}{2}\mathbb{E}\|\nabla f(x_t)\|^2 \leqslant \left(1 + 2AL\eta^2\right)\left(\mathbb{E} f(x_t) - f^*\right) - \left(\mathbb{E} f(x_{t+1}) - f^*\right)$$

$$+ \eta^2 LB^2 + \eta^2 L\mathbb{E}\mathsf{L}_t,$$

where $A$ and $B$ are given in Lemma C.6. $\qquad\square$

The following lemma (Lemma C.8) additionally handles the variance of two-point gradient estimator $\mathsf{L}_t := \|\hat{\nabla} f(x_t; \xi_t, v_t) - \nabla f(x_t; \xi_t)\|^2$ using the upper bound from Theorem 2.2 in the per-iteration recursion.

**Lemma C.8.** *Suppose Assumption 2.1 is satisfied. Under the same setting as Lemma C.7, if the learning rate $\eta \leqslant \frac{1}{2L}$, then*

$$\frac{\delta\eta}{2}\mathbb{E}\|\nabla f(x_t)\|^2 \leqslant \left(1 + 4L^2\eta^2 + 2L^2\beta_V\eta^2\right)\left(\mathbb{E} f(x_t) - f^*\right) - \left(\mathbb{E} f(x_{t+1}) - f^*\right) \qquad (11)$$

$$+ \eta^2 LB^2(1 + \beta_V) + \eta^2\mu^2\alpha_V,$$

*where $\rho_V := \mathbb{E}\|v\|^4 - \delta^2 d^2$, $\alpha_V := L^3\mathbb{E}\|v\|^4$, $\beta_V := 2\delta^2 d + \rho_V + 1 - 2\delta$, and $B^2 := 2L\left(f^* - \mathbb{E}_{\xi\sim\Xi} f_\xi^*\right)$.*

*Proof.* By Theorem 2.2, we have for any $a \in \mathbb{R}^d$,

$$\mathbb{E}_{v\sim V} a^\top(vv^\top)^2 a \leqslant \delta^2 d\|a\|^2 + \frac{\|a\|^2}{2}\rho_V + \frac{\|a\|^2}{2}\sqrt{\rho_V^2 + 4\delta^2(d-1)\rho_V}$$

$$\leqslant 2\delta^2 d\|a\|^2 + \rho_V\|a\|^2$$
$$= \left(2\delta^2 d + \rho_V\right)\|a\|^2,$$

where $\rho_V := \mathbb{E}\|v\|^4 - \delta^2 d^2$. Then

$$\mathbb{E}\|\hat{\nabla}f(x;v) - \nabla f(x)\|^2 \leqslant \nabla f(x)^\top \left(vv^\top\right)^2 \nabla f(x) + (1-2\delta)\|\nabla f(x)\|^2 + L^2\mu^2\mathbb{E}\|v\|^4$$
$$\leqslant \left(2\delta^2 d + \rho_V + 1 - 2\delta\right)\|\nabla f(x)\|^2 + L^2\mu^2\mathbb{E}\|v\|^4.$$

Similarly, we obtain

$$\mathbb{E}\|\hat{\nabla}f(x;v,\xi) - \nabla f(x,\xi)\|^2 \leqslant \left(2\delta^2 d + \rho_V + 1 - 2\delta\right)\|\nabla f(x,\xi)\|^2 + L^2\mu^2\mathbb{E}\|v\|^4$$
$$\leqslant 2L\left(2\delta^2 d + \rho_V + 1 - 2\delta\right)\mathbb{E}\left(f(x;\xi) - f^* + f^* - f_\xi^*\right) + L^2\mu^2\mathbb{E}\|v\|^4$$
$$= 2L\left(2\delta^2 d + \rho_V + 1 - 2\delta\right)\mathbb{E}\left(f(x) - f^*\right) + L^2\mu^2\mathbb{E}\|v\|^4$$
$$+ B^2\left(2\delta^2 d + \rho_V + 1 - 2\delta\right).$$

By Lemma C.7, we have

$$\frac{\delta\eta}{2}\mathbb{E}\|\nabla f(x_t)\|^2 \leqslant \left(1 + 2AL\eta^2\right)\left(\mathbb{E}f(x_t) - f^*\right) - \left(\mathbb{E}f(x_{t+1}) - f^*\right)$$
$$+ \eta^2 LB^2 + \eta^2 L\mathbb{E}\mathsf{L}_t,$$

where $A$ and $B$ are given in Lemma C.6, and $\mathsf{L}_t := \|\hat{\nabla}f(x_t;\xi_t,v_t) - \nabla f(x_t;\xi_t)\|^2$. For notional convenience, we define $\beta_V := 2\delta^2 d + \rho_V + 1 - 2\delta$. Combining both upper bounds, we obtain,

$$\frac{\delta\eta}{2}\mathbb{E}\|\nabla f(x_t)\|^2 \leqslant \left(1 + 2AL\eta^2\right)\left(\mathbb{E}f(x_t) - f^*\right) - \left(\mathbb{E}f(x_{t+1}) - f^*\right)$$
$$+ \eta^2 LB^2 + \eta^2 L\left[2L\beta_V\mathbb{E}\left(f(x) - f^*\right) + L^2\mu^2\mathbb{E}\|v\|^4 + B^2\beta_V\right]$$
$$\leqslant \left(1 + 4L^2\eta^2 + 2L^2\beta_V\eta^2\right)\left(\mathbb{E}f(x_t) - f^*\right) - \left(\mathbb{E}f(x_{t+1}) - f^*\right)$$
$$+ \eta^2 LB^2(1 + \beta_V) + \eta^2 L^3\mu^2\mathbb{E}\|v\|^4$$

It completes the proof. $\qquad\square$

Evaluating the bias of the two-point random smoothing estimator (Eq. (2)) only requires the gradient $\nabla f(\cdot;\xi) : \mathbb{R}^d \to \mathbb{R}$ to be locally Lipschitz continuous. Then we will apply the following Taylor theorem (Hiriart-Urruty et al., 1984; Luc, 1995):

**Lemma C.9** (Taylor's Theorem). *If the function $f : \mathbb{R}^d \to \mathbb{R}$ is continuously differentiable and has locally Lipschitz gradient, then there exists $c \in ]x,y[ \subset \mathbb{R}^d$ and $M_c \in D^2 f(c)$ such that*

$$f(x) - f(y) = \langle\nabla f(y), x - y\rangle + \frac{1}{2}\langle M_c(x-y), x-y\rangle.$$

*Here, $]x,y[$ represents the open rectangular defined by $x, y \in \mathbb{R}^d$.*

*Remark.* Using the $L$-smoothness of $f$, we can show that there exists the upper bound

$$f(x + \mu v) - f(x) \leqslant \mu v^\top\nabla f(x) + L\mu^2\|v\|^2.$$

It leads to the approximation of the zeroth-order gradient estimation for $\nabla f(x)$:

$$\hat{\nabla}f(x;v) \approx \frac{v}{\mu}\Big[f(x + \mu v) - f(x)\Big] = vv^\top\nabla f(x) + L\mu\|v\|^2 v.$$

Therefore, the Mean-Squared Error (MSE) of $\hat{\nabla}f(x;v)$ is bounded by

$$\mathbb{E}\|\hat{\nabla}f(x;v) - \nabla f(x)\|^2 \leqslant \nabla f(x)^\top\left(vv^\top\right)^2\nabla f(x) + (1-2\delta)\|\nabla f(x)\|^2 + L^2\mu^2\mathbb{E}\|v\|^4.$$

## D   DETAILS ON EXPERIMENTS SETUP

This section outlines the information for replicating our experiments. All source codes, including visualization scripts, are provided with our submission.

**Hardware and System Environment**   We conducted our experiments on a cluster running RHEL8, equipped with Dual AMD EPYC 9124 processors and eight NVIDIA RTX 6000 Ada Generation graphics cards. Our code was tested using Python version 3.10.10. Additional dependencies are specified in the supplementary 'requirements.txt' file.

**Hyperparameters**   For the LLM fine-tuning task, we employed the following hyperparameters:

- Learning rate $\eta$: $10^{-4}$;
- Perturbation size $\mu$: $10^{-5}$;
- Zeroth-order gradient estimation batch size $b$: 2;
- Stochastic gradient updates batch size: 16.

## E   ADDITIONAL EXPERIMENTS

In this section, we include additional experiments figures that are not presented in the main text.

### E.1   COMPARISON WITH OTHER RANDOM PERTURBATIONS

In this subsection, we additionally consider the Rademacher perturbation and the random coordinate perturbation in fine tuning the language model. To avoid the significant overlapping in their training loss curves, we also present the boxplot of their training loss within the last 5000 steps for better comparison.

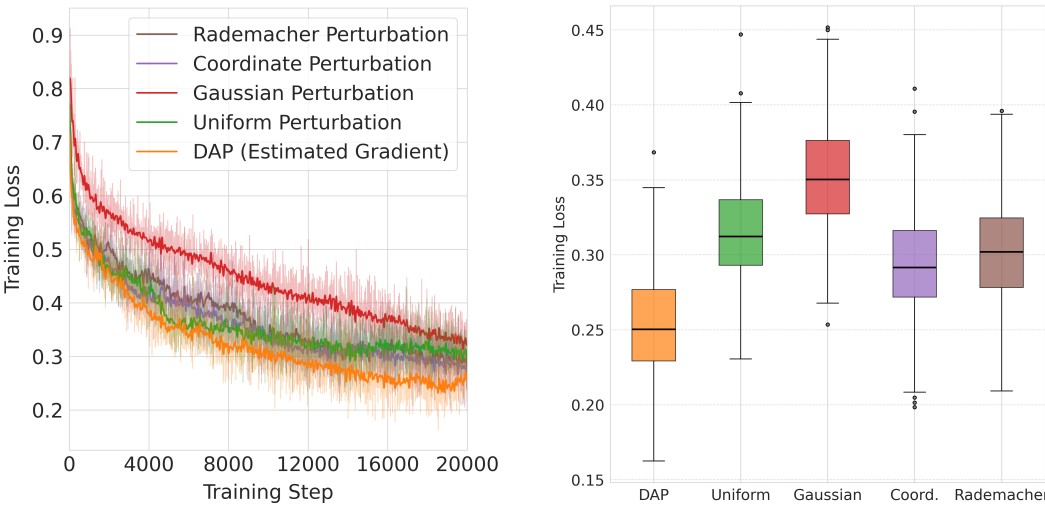

Figure 5: Performance comparison of different optimization methods for fine-tuning OPT-1.3b on SST-2. Compared to Figure 4, we additionally include other two constant magnitude perturbations: Rademacher perturbation and Random coordinate perturbation. The left figure presents the boxplot of training losses within the last 5000 steps.

### E.2   THE ADDITIONAL PRACTICAL APPLICATION: MESH OPTIMIZATION

In this subsection, we consider the mesh optimization problem (Hoppe et al., 1993), which serves as another application of the zeroth-order optimization with utilizing our proposed DAPs. To find the solution of the partial differentiable equations (PDEs), it commonly applies the numerical method such as the finite volume method (Eymard et al., 2000). Such method requires a pre-defined mesh to describe the critical points on the solution surface and the boundary condition. The mesh optimization problem is optimizing the pre-defined mesh to make the PDE solver have better performance by adjusting vertex positions, element connectivity, and local mesh density to minimize numerical errors and improve solution accuracy.

In this experiment, we consider the mesh optimization problem of 2D Poisson's equation (Evans, 2022):

$$\Delta\varphi = f,$$

where $\varphi : \mathbb{R}^2 \to \mathbb{R}$ and $f : \mathbb{R}^2 \to \mathbb{R}$ are both twice differentiable continuous functions and $\Delta$ is the Laplace operator (i.e. $\Delta f(x_1, x_2, \ldots, x_n) = \sum_{i=1}^{n} \frac{\partial^2 f}{\partial x_i^2}$). It is commonly used to model the pressure field of the incompressible Navier-Stokes equation (Acheson, 1990). Its solution is illustrated in Figure 6; our goal is to optimize the coarse mesh to make the numerical solution over the given coarse mesh closer to the solution over the the high-resolution fine mesh.

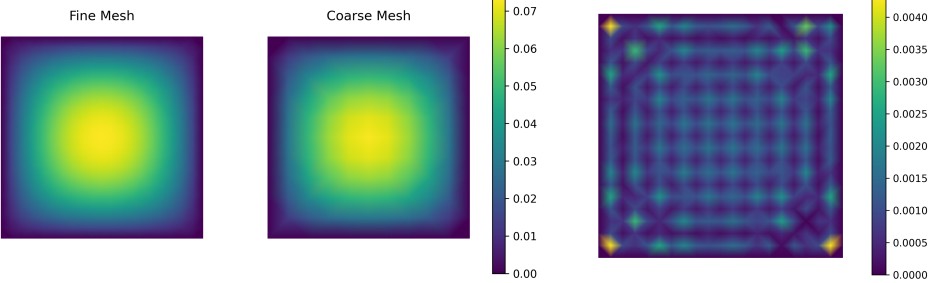

Figure 6: Numerical solutions of the Poisson equation $\Delta\varphi = 1$ with Dirichlet boundary conditions. The left panel compares two numerical solutions: one computed on a fine mesh ($20 \times 20$) and another on a coarse mesh ($10 \times 10$). The right panel shows the difference between these solutions. The discrepancy is particularly pronounced at the corners, suggesting that denser vertices locating may be necessary in these regions for better accuracy.

The following figure applies the zeroth-order optimization method to minimize the $L_\infty$ distance (i.e. the max absolute error) between the interpreted numerical solution over the coarse mesh and the numerical solution over the fine mesh. The DAP method achieves better performance than the baseline approaches (including the uniform perturbation and the Gaussian perturbation).

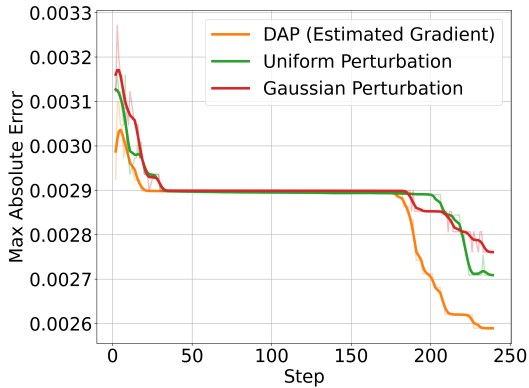

Figure 7: The training loss of the mesh optimization problem for minimizing the $L_\infty$ distance (i.e. the max absolute error) between the interpreted numerical solution over the coarse mesh and the numerical solution over the fine mesh. We use the batch size $b = 512$, the perturbation step $\mu = 10^{-5}$, and the learning rate $\eta = 0.1$.

We also visualize the estimated gradient at the beginning of the training. We use the two-point gradient estimator with the batch size $b = 100000$ and $\mu = 10^{-5}$ to obtain the estimated ground truth gradient. As shown in Figure 8 (Left), only the four corners of the mesh have significant weights. In this case, the anisotropy nature of the DAP leads to more accurate estimation on important vertices.

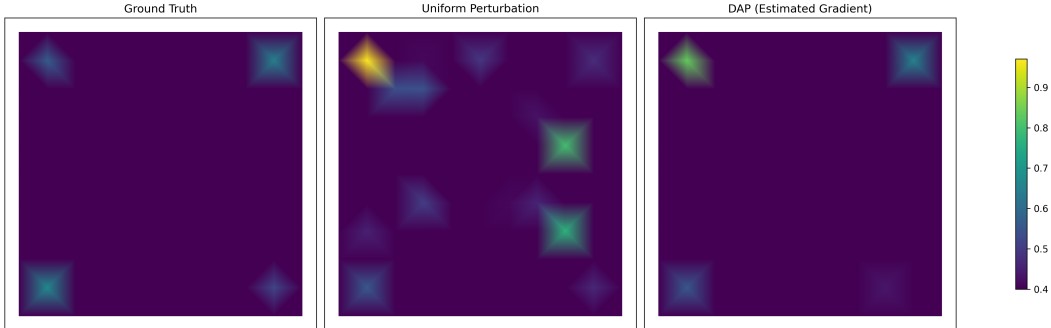

Figure 8: The illustration of the estimated gradient using the uniform perturbation and the DAP with the magnitude larger than $0.4$. The batch size of obtaining the ground truth is $b = 100000$. The batch size for obtaining the estimated gradient is $b = 100$.

## F    DISCUSSIONS ON THE ACCUMULATIVE ERRORS

When approximating the stochastic gradient $\nabla f(x; \xi)$, the two-point gradient estimation naturally introduces additional approximation error. We have previously characterized this error term in the one-step improvement analysis:

$$\eta^2 \mathbb{E} \|\hat{\nabla} f(x; v) - \nabla f(x)\|^2 \leqslant \eta^2 \left[ \nabla f(x)^\top \left(vv^\top\right)^2 \nabla f(x) + (1 - 2\delta)\|\nabla f(x)\|^2 + L^2\mu^2\mathbb{E}\|v\|^4 \right]$$

$$\leqslant \eta^2 \left(2\delta^2 d + \rho_V + 1 - 2\delta\right) \|\nabla f(x)\|^2 + \eta^2 L^2\mu^2\mathbb{E}\|v\|^4.$$

We note that the gradient term $\|\nabla f(x)\|^2$ will be merged together in the convergence analysis we have built in Theorem C.3 and Theorem C.4. Our main focus falls into the last error term $L^2\mu^2\mathbb{E}\|v\|^4$. When telescoping the one-step iteration from Eq. (11), we obtain

$$\text{Accumulative Errors} = \frac{2}{c\delta}\eta\mu^2\alpha_V$$

for the strongly convex objective functions, where $c$ is the strongly convex constant, $\eta$ is the learning rate, and $\alpha_V = L^3\mathbb{E}\|v\|^4$; and

$$\text{Accumulative Errors} = \frac{2\eta}{\delta}\mu^2\alpha_V$$

for the non-convex objective function, where $\eta$ and $\alpha_V$ are defined as above. Both error terms have been reflected in the convergence upper bound and their dependence on $\mu^2$ is common in classical zeroth-order optimization literature (Kozak et al., 2023).

## G    LIMITATIONS

While our theoretical and empirical results demonstrate the potential of DAPs, there are aspects that warrant further investigation. First, when the dimension $d$ is extremely large, the projection steps in sampling DAPs requires storing the full gradient estimates, which introduces additional memory overhead compared to simpler perturbation schemes like uniform or Gaussian perturbations. Second, while DAPs show advantages in scenarios with sparse gradients, they may not perform as well when dealing with dense gradients, as the error introduced by gradient estimation could potentially outweigh the benefits of directional alignment. Finally, although our experiments demonstrate the effectiveness of DAPs on both synthetic problems and language model fine-tuning, we have not extensively tested the method across a broader range of applications and problem settings, and the performance benefits may not generalize to all optimization scenarios.

