# OpenReview forum: "Revisiting Zeroth-Order Optimization:  Minimum-Variance Two-Point Estimators and  Directionally Aligned Perturbations"
_ICLR.cc/2025/Conference — ICLR 2025 Spotlight_

### Official Review · Reviewer_aCKy · 2024-10-23

**Soundness:** 3
**Presentation:** 3
**Contribution:** 2
**Rating:** 8
**Confidence:** 3

**Summary:**

In this paper, the author(s) propose a new perspective on zeroth-order methods, focusing on an optimization problem that minimizes the variance. After carefully crafting the problem instance, based on constraints that are standard in literature, some comments on classic choices of fixed-length perturbations are made. This is the right motivation to proceed and advocate for  "directionally aligned perturbations", the other "optimal choice". In particular, the theorems derived in this branch for convergence of SGD under standard assumptions make explicit the role of higher moments of the distribution, while still recovering nice to parse upper bounds. To conclude, the author(s) come back to the directionally aligned perturbations and show some experiments for synthetic datasets and a Language task. The practical feasibility of their optimization problem is also discussed.

**Strengths:**

The paper presents a nice idea to describe a framework. The optimization problem set up is easily solvable, making the narrative flow. The result is approachable, clear, allowing to derive convergence rates that are expressive of all the dependencies on the various parameters. The plot converges towards "defending" these directionally aligned perturbations, which are of interest even simply because the standard choice is the other optimal one. By passing through simple examples, they are also able to experimentally verify their formulation. Overall a very standard but structured paper in optimization.

**Weaknesses:**

See also the questions below.

The main weakness is about wording: you claim you identify the optimal distribution in the abstract, while indeed you do not. I feel like the sentence should be adjusted. You find a sufficient condition for optimality subject to a construction of isotropic perturbations (you $\delta$-unbiasedness) and a Taylor approximation. This to me is not identifying an optimal distribution, nor identifying anything optimal at all, if not only wrt your specific criterion, which then you would have to specify anyway.

I feel like the other main weakness is experimental validation. I believe you have brought the best results you could find, and still, the improvement is marginal. For example, figure $3$ right is really an improvement in machine precision. Figure $4$ is more promising. I also acknowledge that we should not care about SOTA but about understanding, so this is a weakness that is not suggesting any further comment.

The other point is that you do not discuss the accumulation of errors when you (i)
 perform the Taylor approximation and (ii) perform the gradient estimation. I believe the two should be theoretically explored further to understand in restricted settings how much is lost wrt the convergence theorems.

Lastly, no limitations are discussed.


###### Typos
Please do not count these as weaknesses.
- You never define $\nabla f(x; \xi, v)$, (e.g. line 085), since the notation for derivatives variable, I would define it.
- "In the mean while" (line 166), meanwhile;
- The numbering of lists as $(1), (2)$ etc resembles a lot equations, not a typo but a potential source of anti-dynamic reading;
- line 215, you say $< 0$, probably meant to be finite.
- line 264 "to a specific types of..."
- line 269 "achiving"
- "classitcal" (line 472)
- "it solves the projection is..." (line 864)
- corollary 3.2 (b) the sentence is not correct logically. If we add the assumption by choosing further specific scalings we get the result, right?

**Questions:**

1. Have you analyzed the impact of the Taylor approximation in your analysis?
2. Have you analyzed the impact of the difference between theoretical analysis and estimating gradients in the wild? I acknowledge the experiment for the easy functions, where it is exact.
3. Why would the assumption of isotropic noise $\mathbb{E}[vv^\top] = \delta I_d$ be valuable in terms of perturbations? Why not something else?



I am very open to changing my score once the issues I have raised are addressed!

---

> ### Author Response · Authors · 2024-11-21
> **Responses (1/2)**
>
> We sincerely thank you for the thorough and constructive feedback. Below we respond to the comments in Weaknesses (W) and Questions (Q). We have also uploaded a Paper Revision with modified content in red.
>
> * **W1**: Unspecified criterion for the word "optimal".
>
>   **A**: We thank the reviewer for highlighting this ambiguity. We acknowledge that our use of the term "optimal" was imprecise in this context. We have revised the manuscript to replace "optimal perturbation" with a more specific description: achieving minimum asymptotic variance as the perturbation step $\mu\to 0$. This is also what our work aims to convey: The widely used fixed length random perturbation is not the only distribution that achieves such minimum variance; the DAP presents an alternative approach to achieve the same criterion.
>
> * **W2**: Experiment Validation.
>
>   **A**: We appreciate the reviewer's valuable feedback. To address this concern, we have conducted an additional experiment focusing on mesh optimization—a crucial component in numerical PDE solving. This experiment aims to optimize vertex positions in pre-defined meshes to enhance PDE solver performance by minimizing numerical errors and improving solution accuracy.
>
>   The results, detailed in Appendix E (Additional Experiments), demonstrate that DAP significantly outperforms both uniform and Gaussian perturbation methods (Figure 7). This experiment serves two purposes: it not only showcases DAP's superior performance over classical random perturbations but also illustrates the anisotropic behavior in gradient estimation in a practical application—a key distinguishing feature of DAP compared to conventional random perturbation methods.
>
>   In Appendix E.2, Figure 6 compares numerical PDE solutions between coarse and fine meshes, revealing that mesh corners require denser vertex positioning for improved accuracy. Furthermore, Figure 8 visualizes the gradient estimation norms using uniform distribution versus DAP, demonstrating that DAP's gradient estimator structure more closely aligns with the ground truth and appropriately emphasizes the mesh corners.
>
> * **W3 & Q1**: Impact of Taylor approximation in the theoretical analysis.
>
>   **A**: We thank the reviewer for pointing out this critical point. In our revised version, we have added an additional remark and an appendix section to discuss the error caused by the Taylor approximation. We shortly summarize it as follows: We use the bounded Hessian assumption to upper bound the residual term of the Taylor series. When the perturbation step $\mu$ is sufficiently small, we will be able to control the error of this error term.
>
>   We note that although we didn't explicitly state this error term in our main text, we have already considered its impact in our original convergence analysis (Lemma C.8). We have further discussed its influence in the final upper bound in Appendix G Discussions on the Accumulative Errors.
>
> * **W4**: Discussions on the limitation.
>
>   **A**: We appreciate the reviewer's suggestion. We have added the limitation section in the revised paper. Shortly speaking, we have emphasis the following three points: (1) The projection step requires to store the whole gradient estimation, which may cost too much memory. (2) The additional estimation step in Algorithm 2 introduces additional error in the gradient estimation, which potentially makes the DAP underperform standard methods in traditional tasks. (3) More experiments could be needed to clearly verify when the DAP method is better and when the DAP method is not.

---

> > ### Author Response · Authors · 2024-11-21
> > **Responses (2/2)**
> >
> > * **Q2**: Difference between theoretical analysis and estimating gradients in the wild.
> >
> >   **A**: We appreciate the reviewer's constructive comment. We add one additional experiment to better illustrate the gradient estimation using the DAP in the practical mesh optimization problem as shown in Appendix E Additional Experiments. The Figure 8 reflects the same phenomenon as we have presented in Figure 2, which we have demonstrated in our response to W2.
> >
> > * **Q3**: Assumption of $E[vv^\top] = \delta I_d$.
> >
> >   **A**:  This assumption is based on the asymptotic unbiasedness of the two-point gradient estimator as $\mu\to 0$. In the limiting case,  we always have $\lim_{\mu\to 0}E[\hat{\nabla}f(x;v)]=\delta \nabla f(x)$, which ensures that the estimated gradient in expectation has the same direction as the true gradient (up to a scale constant $\delta$). This property is crucial for the convergence analysis of SGD.
> >
> >   We follow this standard setting instead of other settings because it has been widely considered in existing work including [Kozak2023]'s Equation (P.2) and [Rando2024]'s Assumption 2 and it has covered sufficient amount of perturbation distributions.
> >
> >   [Kozak2023] Kozak, David, et al. "Zeroth-order optimization with orthogonal random directions." *Mathematical Programming* 199.1 (2023): 1179-1219.
> >
> >   [Rando2024] Rando, Marco, et al. "Stochastic zeroth order descent with structured directions." *Computational Optimization and Applications* (2024): 1-37.
> >
> > **Typos**: Thanks for the careful reading. We have thoroughly revised the paper to fix these typos and to improve the overall clarity.

---

> ### Comment · Area_Chair_tACs · 2024-11-25
>
> Dear Reviewer aCKy,
>
> The author discussion phase will be ending soon. The authors have provided detailed responses. Could you please reply to the authors with whether they have addressed your concern and whether you will keep or modify your assessment on this submission?
>
> Thanks.
>
> Area Chair

---

> ### Comment · Reviewer_aCKy · 2024-11-25
> **Score Update**
>
> Dear author(s),
> thank you for your detailed response. I have waited for further engagement by the other authors because I thought I had nothing to add. On my side, it looks like you have sufficiently addressed my comments.
>
> I will raise the score, and vouch for acceptance. Please note that since there is no "7" and "6" is a marginal grade (which I want to avoid), my score is 8 on this scale.
>
> Good luck.

---

### Official Review · Reviewer_gWHS · 2024-10-29

**Soundness:** 3
**Presentation:** 4
**Contribution:** 3
**Rating:** 6
**Confidence:** 3

**Summary:**

The paper studies the zeroth-order gradient estimator and identifies the optimal distribution of random perturbations that minimize the gradient estimator's variance.
The problem is formulated as a constrained optimization problem. And it is shown that the optimal perturbations maintain a fixed length or align directionally with true gradient.
These gives two classes of random perturbations that achieve the minimum variance : Constant magnitude perturbations and Directionally aligned perturbations.
Convergence of SGD with both these classes of perturbations are proved. And some experimental results are shown.

**Strengths:**

The problem studied is of significant interest to the optimization community. And it shows two classes of random perturbations that give minimum variance.

**Weaknesses:**

In the main theorem (Theorem 2.2), what about only if part ? Does it happen that equality holds in theorem only if the given conditions (a) or (b) is satisfied ?
A discussion of this would be interesting.
The experimental results are weak. Only one practical application of language model optimization is given.
No comparisons with other constant magnitude perturbations: random coordinate/direction sampling and Rademacher distribution are shown.
Why DAP perturbations give better performance than uniform perturbation in experiments is not clear. As the theorem says that theoretically both give minimum variance.

**Questions:**

See weakness.

---

> ### Author Response · Authors · 2024-11-21
>
> We sincerely thank your careful reading and constructive feedback. Below we respond to the comments in two Weaknesses (W). We have also uploaded a Paper Revision with modified content in red.
>
> * **W1**: Regarding the "only if" part of Theorem 2.2.
>
>   **A**: We appreciate this insightful question. While we establish a sufficient condition for achieving minimum variance, we acknowledge that this condition is not necessary. Finding the necessary condition (of achieving the minimum variance) is much harder than finding the sufficient condition. We can easily construct a mixed distribution that is neither the DAP nor the constant magnitude perturbation by taking the probability $p$ to be the DAP and the probability $1-p$ to be the constant magnitude perturbation.
>
>   We believe it is possible to obtain the necessary and sufficient condition by further restricting the distribution class of the random perturbations. However, we still choose to present our current version due to the following reasons: (1) Our chosen distribution class  (i.e. all distribution $V$ such that $E_{v\sim V} vv^\top = \delta I_d$) aligns with standard frameworks in the field, such as [Kozak2023] and [Rando2024]. While restricting this class might yield necessary and sufficient conditions, it would deviate from established conventions. (2) The directional alignment condition we present introduces a novel class of random perturbations with compelling anisotropic properties. We believe this discovery makes a meaningful contribution to the optimization community, even without characterizing the complete set of minimum-variance distributions.
>
>   [Kozak2023] Kozak, David, et al. "Zeroth-order optimization with orthogonal random directions." *Mathematical Programming* 199.1 (2023): 1179-1219.
>
>   [Rando2024] Rando, Marco, et al. "Stochastic zeroth order descent with structured directions." *Computational Optimization and Applications* (2024): 1-37.
>
> * **W2**: Experiments.
>
>   * Other practical application.
>   * Comparison with other constant magnitude perturbations.
>   * Explain why DAP is better (even they have the same variance).
>
>   **A**: Thank you for these constructive suggestions. We have expanded our experimental section in Appendix E with two key additions: (1) A new practical application involving mesh optimization for PDE solvers, where we optimize mesh configurations to enhance numerical accuracy. (2) The comparison among other constant-magnitude perturbation in the language model fine-tuning experiment.
>
>   The reason why the DAP achieves better performance on the certain application is that the DAP will automatically detect which direction is more important; therefore, it provides more accurate estimation on these directions. We have illustrated this phenomenon in the Figure 2.
>
>   To explain this phenomenon in a more practical example, we further illustrate it in the mesh optimization problem. In Appendix E.2, Figure 6 compares numerical PDE solutions between coarse and fine meshes, revealing that mesh corners require denser vertex positioning for improved accuracy. Furthermore, Figure 8 visualizes the gradient estimation norms using uniform distribution versus DAP, demonstrating that DAP's gradient estimator structure more closely aligns with the ground truth and appropriately emphasizes the mesh corners. As the result, higher accuracy on these corner vertices improve the performance of the DAP.

---

> > ### Comment · Reviewer_gWHS · 2024-11-25
> >
> > One of my comments about experimental validation has been more or less addressed. But the other remains. No discussion or insights has been given about the only if condition.

---

> > > ### Author Response · Authors · 2024-11-25
> > >
> > > We sincerely thank the reviewer for these constructive suggestions, which clearly helped us strengthen our paper. We will add the following paragraph (adapted from our previous rebuttal) to our revised paper right after Theorem 2.2. It discusses on the construction of perturbation that is neither the DAP or the fixed length, and further comments on the potential senario where the sufficient and necessary condition could be derived. We sincerely hope the reviewer find this modification useful in improving our manuscript:
> > >
> > > **On the Necessity of Minimum Variance Conditions** In our previous theorem (Theorem 2.2), we only present the sufficient condition of achieving the asymptotic minimum variance as the perturbation step $\mu$ tends to $0$.  A simple counterexample demonstrates that this condition may not be necessary: consider a mixed distribution that takes the DAP with probability $p$ and the uniform perturbation over the sphere with probability $1-p$. Such a distribution would also achieve minimum variance while satisfying neither condition exclusively. Extending the condition to sufficient and necessary condition would be an interesting but challenging topic. Here we present one potential senario where we may obtain the necessary and sufficient condition: In the one-dimensional case, assuming the random perturbation satisfying $E v = 0$ and $E v^2 = 1$, the unique distribution of achieving the minimum variance is the Rademacher distribution, which is naturally derived by considering the Taylor expansion. This case may further be extended to $d$-dimension with requiring all entries in the random perturbation to be mutually independent; however, this extension would be out of the scope of our paper and excludes many interesting random distributions. We will still stick to our $\delta$-unbiased perturbations (Assumption 2.1) in the remaining of our manuscript.

---

> > > > ### Comment · Reviewer_gWHS · 2024-11-25
> > > >
> > > > I've updated my score based on the modifications.

---

> ### Comment · Area_Chair_tACs · 2024-11-25
>
> Dear Reviewer gWHS,
>
> The author discussion phase will be ending soon. The authors have provided detailed responses. Could you please reply to the authors with whether they have addressed your concern and whether you will keep or modify your assessment on this submission?
>
> Thanks.
>
> Area Chair

---

### Official Review · Reviewer_F997 · 2024-11-01

**Soundness:** 3
**Presentation:** 3
**Contribution:** 3
**Rating:** 6
**Confidence:** 3

**Summary:**

This paper explores the two-point zeroth-order gradient estimator and identify the optimal distribution of random perturbations that minimizes the estimator's variance. This paper formulates it as a constrained functional optimization problem over the space of perturbation distributions. This paper reveals that optimal perturbations either maintain a fixed length or align directionally with the true gradient. While existing research has largely focused on fixed-length perturbations, the potential advantages of directional alignment have been overlooked. To address this gap, this paper delves into the theoretical and empirical properties of the directionally aligned perturbation (DAP) scheme, which adaptively offers higher accuracy along critical directions. Additionally, this paper provides a convergence analysis for stochastic gradient descent using $\delta$-unbiased random perturbations, extending optimal complexity bounds to a wider range of perturbations. Through empirical evaluations on both synthetic problems and practical tasks, we demonstrate that DAPs outperform traditional methods under specific conditions.

**Strengths:**

This paper explores the two-point zeroth-order gradient estimator and identify the optimal distribution of random perturbations that minimizes the estimator's variance. This paper formulates it as a constrained functional optimization problem over the space of perturbation distributions. This paper reveals that optimal perturbations either maintain a fixed length or align directionally with the true gradient. While existing research has largely focused on fixed-length perturbations, the potential advantages of directional alignment have been overlooked. To address this gap, this paper delves into the theoretical and empirical properties of the directionally aligned perturbation (DAP) scheme, which adaptively offers higher accuracy along critical directions. Additionally, this paper provides a convergence analysis for stochastic gradient descent using $\delta$-unbiased random perturbations, extending optimal complexity bounds to a wider range of perturbations. Through empirical evaluations on both synthetic problems and practical tasks, we demonstrate that DAPs outperform traditional methods under specific conditions.

**Weaknesses:**

I don't think the study over problem Eq. (3) is meaningful.
Under the theory of this paper, the random coordinate is better than Gaussian random vector.
However, just as pointed out in Theorem 1 of  "Fine-tuning language models with just forward passes'', the Gaussian random vector can provide a "Dimension-Free Rate''.
Unfortunately, the random coordinate can not  guarantee this ``Dimension-Free Rate'' even it is good under the thooery of this paper.

The experiments in this paper do not show significant advantages of DAP over other estimations.

**Questions:**

No.

---

> ### Author Response · Authors · 2024-11-21
>
> We sincerely thank you for the comment. Below we respond to the comments in two Weaknesses (W). We have also uploaded a Paper Revision with modified content in red.
>
> * **W1** (Dimension-Free Rate): I don't think the study over problem Eq. (3) is meaningful. Under the theory of this paper, the random coordinate is better than Gaussian random vector. However, just as pointed out in Theorem 1 of "Fine-tuning language models with just forward passes'', the Gaussian random vector can provide a "Dimension-Free Rate''. Unfortunately, the random coordinate can not guarantee this ``Dimension-Free Rate'' even it is good under the theory of this paper.
>
>   **A**: We politely disagree with this point. We would like to highlight that the dimension-free rate in the paper "*Fine-tuning language models with just forward passes*" requires the Local $r$-Effective Rank assumption (the Assumption 1 in their paper). The dimension-free rate in that paper is not because the Gaussian perturbation is used; instead, it is because the local $r$-effective rank assumption is made. Without adding additional assumption, the dependence on the dimension $d$ is not avoidable (see [Duchi2015]).
>
>   [Duchi2015] Duchi, John C., et al. "Optimal rates for zero-order convex optimization: The power of two function evaluations." *IEEE Transactions on Information Theory* 61.5 (2015): 2788-2806.
>
> * **W2** (No significant advantages of DAP): The experiments in this paper do not show significant advantages of DAP over other estimations.
>
>   **A**: We appreciate the reviewer's comments and would like to highlight: First, Figure 3 demonstrates that the DAP achieves significant improvements in estimating the gradient of the product function with respect to the $\tau$-effective MSE. Furthermore, as shown in Figure 4, the convergence performance using DAPs is substantially faster compared to both other methods.
>
>   To directly address the reviewer's concerns, we have conducted an additional experiment on the mesh optimization problem. The results, presented in Figure 7, further validate that DAP outperforms the other two random perturbations in this context as well.
>
>   We would like to clarify that we have not claimed our method universally outperforms traditional approaches across all scenarios, nor was this our research objective. Instead, our primary contribution lies in identifying and characterizing a novel type of random perturbation. The anisotropic nature of DAP is significantly different from existing random perturbation methods, marking it possible to achieve particular advancement in specific fields. We respectfully ask the reviewer to consider this novel finding and the important theoretical contribution when evaluating our work.

---

> > ### Comment · Reviewer_F997 · 2024-11-21
> >
> > Thank you for the replies. However, the  dimension-free rate in that paper is  because the Gaussian perturbation is used and the local-effective rank assumption is made. Instead, if taking the random coordinate, a $d$ dimension dependence will hold. However, the random coordinate is optimal in your theory.
> > Of course, you can also emprically show me that zeroth-ofder algorithm with the random coordinate direction can achieve fast convergence rate the same as the Gaussian in LLM fine-tune.

---

> > > ### Author Response · Authors · 2024-11-21
> > >
> > > We appreciate the reviewer's recognition that the dimension-free rate in the referenced paper requires the local-effective rank assumption. We also thank the reviewer for pointing it out that even under the assumption of $r$-local effective rank, there exist some perturbations (e.g. the random coordinate) that cannot achieve the dimension-free rate.
> > >
> > > We agree with this point but it doesn't mean studying the variance of two-point gradient estimator is meaningless, because not all problems satisfy the $r$-local effective rank assumption. We would like to clarify two points:
> > >
> > > 1.  Our work considers the standard zeroth-order optimization setting, rather than the case under the $r$-local effective rank assumption. This setting is well-established in the field of zeroth-order optimization, as demonstrated by seminal works like [Ghadimi2013] (1500+ citations) and [Duchi2015] (500+ citations), upon which our research builds. Adding additional asstumptions will definitely improve the dependence on the dimension $d$; however, it also restricts the class of valid optimization problems.
> > >
> > >     * *[Ghadimi2013] Ghadimi, Saeed, and Guanghui Lan. "Stochastic first-and zeroth-order methods for nonconvex stochastic programming." SIAM journal on optimization 23.4 (2013): 2341-2368.*
> > >
> > >     * *[Duchi2015] Duchi, John C., et al. "Optimal rates for zero-order convex optimization: The power of two function evaluations." IEEE Transactions on Information Theory 61.5 (2015): 2788-2806.*
> > >
> > > 2.  In our revised manuscript, we have removed statements comparing Gaussian to minimum-variance perturbations (in Page 4 Constant Magnitude Perturbations and Experiment 5.1 Synthetic Example). We kindly hope this modification improve the focusing of our work. One of our primary goals is to deliver a new type of random perturbation that exhibits anisotropic behavior across different directions, which is new and different from existing classical random perturbations; such anisotropic property has shown its advancement to be applied in extensive fields (as evidenced in our language model fine-tuning and the mesh optimization experiment). If you agree our modification strengthens our paper's focus, we would appreciate if reconsideration of the score could be made.

---

> > > > ### Comment · Reviewer_F997 · 2024-11-24
> > > >
> > > > Thank you for the replies. Though this work does \emph{not} explain the reason why Gaussian or some other distributions can achieve fast convergence rate, this work still has its value. So I raise the score.

---

### Official Review · Reviewer_BjSg · 2024-11-02

**Soundness:** 3
**Presentation:** 3
**Contribution:** 3
**Rating:** 6
**Confidence:** 2

**Summary:**

This paper studies the two-point zeroth-order gradient estimator, specifically focusing on the problem of identifying the optimal distribution of random perturbations that minimizes the estimator's variance. In Section 1, they briefly introduce the preliminary concepts and raise the motivating questions of the work. They first question whether it would be possible to determine the class of optimal distributions of random perturbations in a zeroth-order estimator to minimize its variance, and provide Theorem 2.2 as the answer. In Theorem 2.2, they introduce two sufficient conditions for the question, which are constant magnitude perturbations and a novel condition called directionally aligned perturbations (DAPs). In Section 4, they take a closer look at DAPs and provide a sampling strategy for practical implementation. Finally, in Section 5, they demonstrate the practical effectiveness of DAPs through two experimental setups with a synthetic example and language model optimization.

**Strengths:**

To the best of the reviewer's understanding, the core contributions of this paper are two parts:
- This paper formalized the problem of characterizing the class of optimal distributions of random perturbations in a zeroth-order estimator to minimize its variance and provided sufficient conditions.
- Based on the first contribution, they conceptualize the novel condition which they name DAPs, provide a way to use it practically, and demonstrate the effectiveness by experiment.

The reviewer thinks these are meaningful contributions. The paper also shows that the complexity of SGD with two-point gradient estimation achieves the best-known sample complexity when the perturbation distribution $V$ is chosen to achieve the minimum variance.

Also, the reviewer thinks the writing of the paper is overall nice.

**Weaknesses:**

This can be closer to a question than a weakness, however, the reviewer is confused about the underlying logic and contribution of Section 3. The reviewer may be missing some elementary points, but they are still confused about the sufficient and necessary condition for minimum variance.

- The most confusing point was the relation between the fourth-order moment. In Theorem 2.2, as addressed in the Remark, it seems (2) has minimum variance when equality holds. But is the converse also true? It seems the terms related to the fourth-order moment only appear in the upper bound. If the converse doesn't hold, isn't the finiteness of the fourth-order moment neither a sufficient nor a necessary condition for achieving minimum variance? In this context, is the finiteness of the fourth-order moment an additional assumption (other than minimum variance) imposed to obtain the results in Section 3?

- The reviewer thinks the observation about the influence of the fourth-order moment addressed in the Remark of Theorem 3.1 can be meaningful by itself. However, the reviewer thought the main focus of the paper was the conditions for minimum variance and specifically DAPs. Yet, the first half of Section 3 seems to be just a convergence analysis of SGD, with the assumption of the finiteness of the fourth-order moment. According to the authors' explanation, the proof heavily relies on arguments considered in prior works.

- In short, what is the role of Theorem 3.1 in the overall context of the paper? It seems Theorem 2.2 is used in the proof; is it crucial? The reviewer thinks it would be better to address a quick overview of Section 3 in the overall context of the paper at the beginning of the section. The reviewer felt lost when first reading Section 3.

**Questions:**

- Could you provide the explanation related to the reviewer's questions in the weakness?

- Is (a) and (b) in Theorem 2.2 sufficient conditions for achieving the minimum variance, or are they also necessary conditions?

- It seems the authors claim in the remark about (a) of Theorem 3.1 that a small $\delta$ leads to more gradient updates. However, it appears that Theorem 3.1 provides an upper bound result, so it may not serve as logical evidence for your discussion. Or do you have a lower bound result as well?


**Minor questions:**
- Is the definition of parameters in (a) and (b) of Theorem 3.1 the same? Is there a reason you are repeating them?
- Did you try to write $< \infty$ in line 215?

---

> ### Author Response · Authors · 2024-11-21
>
> We sincerely thank the reviewer for their thorough and constructive feedback. We appreciate the positive assessment of our work. Below we respond to the comments in Weaknesses (W) and Questions (Q). We have also uploaded a Paper Revision with modified content in red.
>
> * **W1.1**: Is the converse of Theorem 2.2 also true? **&** **Q2**: Is (a) and (b) in Theorem 2.2 sufficient conditions for achieving the minimum variance, or are they also necessary conditions?
>
>   **A**: We appreciate this insightful question raised by the reviewer. The converse is unfortunately not true and we only derive the sufficient condition for achieving the minimum variance. Finding the necessary condition (of achieving the minimum variance) is much harder than finding the sufficient condition. We can easily construct a mixed distribution that is neither the DAP nor the constant magnitude perturbation by taking the probability $p$ to be the DAP and the probability $1-p$ to be the constant magnitude perturbation. We believe it is possible to obtain the necessary and sufficient condition by further restricting the distribution class of the random perturbations. However, we still choose to present our current version due to the following reasons: (1) We prefer to consider the current distribution class (i.e. all distribution $V$ such that $E_{v\sim V} vv^\top = \delta I_d$) instead of further restricting it to obtain the necessary and sufficient condition because the current setting is more standard. The similar distribution class can be found in  [Kozak2023] and [Rando2024]. (2) The directional alignment condition has already revealed a new class of random perturbations which presents an attractive anisotropic behavior; we truly believe it is an interesting and impactful result to share with the optimization community.
>
>   [Kozak2023] Kozak, David, et al. "Zeroth-order optimization with orthogonal random directions." *Mathematical Programming* 199.1 (2023): 1179-1219.
>
>   [Rando2024] Rando, Marco, et al. "Stochastic zeroth order descent with structured directions." *Computational Optimization and Applications* (2024): 1-37.
>
> * **W1.2**: How to elaborate the finite fourth-order moment condition in the upper bound? Is it the necessary condition?
>
>   **A**: We appreciate this insightful question raised by the reviewer. The fourth-order moment condition $E_{v\sim V} \| v\|^4 = \delta^2 d^2$ can be treated as another sufficient condition, which we didn't consider in our paper. It is weaker than the fixed-length perturbation (because $\|v\|^2=\delta d$​​ implies this fourth-moment condition); however, it remains unclear if this condition could be the necessary condition. We believe it could be an interesting topic to be further explored in the future.
>
> * **W2 & W3** : What is the contribution of convergence analysis in Section 3? What is the role of Theorem 3.1 in the overall context of the paper?
>
>   **A**: We appreciate the reviewer's constructive suggestion. We have added an overview paragraph at the beginning of the Section 3.
>
>   In existing literature, the approximation error of the gradient estimation $E_{v\sim V}\\| \hat{\nabla} f(x;v) - \nabla f(x) \\|^2$ is commonly tailored to a specific type of random perturbation. For example, [Nesterov2017] only considers the Gaussian random perturbation; the explicit Gaussian density allows the tight characterization of the dependence on the parameter dimension $d$. By applying the upper bound obtained from Theorem 2.2, we are able to build a much more general upper bound with such optimal dependence.
>
>   [Nesterov2017] Nesterov, Yurii, and Vladimir Spokoiny. "Random gradient-free minimization of convex functions." *Foundations of Computational Mathematics* 17.2 (2017): 527-566.
>
> * **Q3**: Theorem 3.1 only provides an upper bound result, which doesn't support the claim made in the remark of Theorem 3.1.
>
>   **A**: We appreciate the reviewer point out this crucial point. We agree that the upper bound result doesn't support the claim that a small $\delta$ leads to more gradient updates. Here, we intended to state the reason why we only consider the case $\delta=1$ in our worst-case complexity analysis. We have revised the comment (a) to reflect our purpose more clearly.
>
> **Minor questions**: Thanks for pointing out these issues. We have removed the redundant definition of parameters in Theorem 3.1 and fixed the typo in line 215 in our revised paper.

---

> > ### Comment · Reviewer_BjSg · 2024-11-26
> >
> > I sincerely thank the authors for their comprehensive and considerate reply to my review. I believe they have sufficiently addressed the questions I raised.

---

### Official Review · Reviewer_6hyB · 2024-11-03

**Soundness:** 3
**Presentation:** 3
**Contribution:** 3
**Rating:** 8
**Confidence:** 5

**Summary:**

***Summary:***
In the paper the authors study (sufficient?) conditions on the  distribution of the sampling directions in order to build a two-point finite difference estimator of the gradient that is at the same time unbiased and with minimal variance. Then they state convergence results for SGD using this kind of estimators (in the non-convex and stronly-convex case), showing that they achieve the optimal complexity in terms of dimension. Finally they focus on DAP (directionally aligned perturbation), a new estimator which satisfies unbiasedness and minimal variance. They design an algorithm to implement it and show promising numerical experiments.

**Strengths:**

***Main comments:***
The review of the literature is complete, the problem is meaningful and relevant, the theoretical results are significant, the proofs are correct. The paper is interesting and well-written, the presentation is both concise and comprehensible.

**Weaknesses:**

The paper is well written but there are some points with imprecise statements.  See the comments in the Questions box.

1) The stochastic optimization setting (in $\xi$) is not needed in the first part of the paper but only for SGD (Section 3), and it creates confusion.

2) P3, Theorem 2.2: the inequalities are clear, but it is not clear to me what you can deduce from them, as they give only a lower- and upper-bound on the quantity you want to minimize.

**Questions:**

1) P3, Theorem 2.2: the inequalities are clear, but it is not clear to me what you can deduce from them, as they give only a lower- and upper-bound on the quantity you want to minimize. Is it true that the variance is minimal if and only if $\rho_V=0$? Or $\rho_V=0$ is only a sufficient condition? Are conditions (a) and (b) sufficient conditions to get $\rho_V=0$? Apparently no, since from (b) you can not get $\rho_V=0$. To me, it is not completely clear the logic of the reasoning neither the statement. This is true especially in connection with the comment on Gaussian Smoothing at P4: does the fact that $\rho_V>0$ imply that Gaussian Smoothing does not achieve minimal variance? From the inequalities of the Theorem you just know that the variance is lower-bounded and upper-bounded by two different quantities...

2) P6, DAP: for the unknown gradient $\nabla f(x)$, you can apply a small batch of perturbations to obtain an estimated gradient. Second level question: with which distribution do you sample $v$ for the estimator of the gradient used in DAP?


More bibliography:

- Cai, Mckenzie, Yin, Zhang: Zeroth-Order Regularized Optimization (ZORO): Approximately Sparse Gradients and Adaptive Sampling; SIAOPT 2022

- Cai, Mckenzie, Yin, Zhang: A One-bit, Comparison-Based Gradient Estimator; ACHA 2022

- Rando, Molinari, Villa, Rosasco: Stochastic Zeroth order Descent with Structured Directions; COAP 2024

- The paper [Rando, Molinari, Villa, Rosasco: An Optimal Structured Zeroth-order Algorithm for Non-smooth Optimization] has been published in NeurIPS 2023

- Akhavan, Chzhen, Pontil, B. Tsybakov: A gradient estimator via L1-randomization for online zero-order optimization with two point feedback; NeurIPS 2022\\

***Minor comments:***

P2: the formula in Contribution 1 is not correct, should be $\nabla f(x;\xi)$ without the $v$

P2: explain why the constraint $\mathbb{E} vv^T = \delta I$ gives the unbiasedness of the gradient approximation (this is true only for $\mu \to 0$); why do you say it is a linear constraint?

P3, L124: first line of the equation is wrong; in the second line, where is the second order term with $M_c(v)$? Explain better the approximation you make...

P4, DPA: highlight that, in the practice of zeroth order optimization, this condition can not be imposed like it is, since $\nabla f(x)$ is not available

P4, L187: $a^Tv=\pm \sqrt{\delta}\|a\|$

P4, L206: $\hat{\nabla}f(x;\xi)$ has not been defined, but only $\hat{\nabla}f(x;\xi, v)$

P4, L210: comment that the quantity $\min_t \|\nabla f(x_t)\|$ is not something you can check in the practice of zeroth order optimization; in particular, you don't know which one is the best iterate accordingly to the criterion $\|\nabla f(x_t)\|$

P4, L210: $f^*$ not defined

P5, L222: $\mathbb{E}_{\xi} f^*_{\xi}$ not defined

P5, L226: say that $c$ is the strong-convexity constant (it appears only in the definition in the appendix)

P5, L238: comment (a) is not clear to me

P5, Corollary 3.2: "If choosing" is not correct (used twice); the bound $\leq \varepsilon$ has constants involved that are omitted

P6, Fig. 1: the caption is not clear to me

---

> ### Author Response · Authors · 2024-11-21
>
> We sincerely thank the reviewer for their thorough and constructive feedback. We appreciate the positive assessment of our work.  Below we respond to the comments in Weaknesses (W) and Questions (Q). We have also uploaded a Paper Revision with modified content in red.
>
> * **W1**:  The stochastic optimization setting is not needed in the first part of the paper but only for SGD (Section 3), and it creates confusion.
>
>   **A**: We appreciate the reviewer's suggestion. We have added additional subsection (Section 1.1 Paper Structure) in the introduction to describe the structure of our paper, which we hope could improve the overall clarity.
>
> * **W2**: How to deduce the result of Theorem 2.2:
>
>   * **Q1**: Is $\rho_V=0$ a sufficient condition?
>   * **Q2**: $\rho_V \neq 0$ doesn't imply Gaussian smoothing is not optimal.
>
>   **A**: We appreciate the your insightful comments. We acknowledge that our statement regarding the Gaussian smoothing is not accurate. As the reviewer point out, our proposed conditions including ($\rho_V=0$) are only the sufficient condition; it is easy to construct a perturbation that is neither the fixed length and the DAP: We can sample a perturbation from the uniform smoothing with the probability $p$, then sample from the DAP with the probability $1-p$. This construction doesn't change the variance but doesn't satisfy either condition.
>
>   Therefore, from our statement, we cannot conclude that the Gaussian smoothing fails to achieve the minimum variance as claimed by simply evaluating the fourth-order moment $E\\| v\\|^4$. Fortunately, our statement is still correct: the matrix $E (vv^\top)^2 $ can be explicitly solved when the distribution $v$ is known as $N(0, I_d)$; it is a diagonal matrix with $E[(vv^\top)^2_{i,j}]=0$ for $i\neq j$ and $E[(vv^\top)^2_{i,j}]=E[v_i^4] + (d-1)=d+2$ for $i=j$. It concludes that asymptotic variance of $\hat{\nabla} f(x;v)$ as $\mu \to 0$ is $(d+1)\| \nabla f(x)\|^2$ which is larger than the minimum asymptotic variance $(d-1)\| \nabla f(x)\|^2$ achieved by the uniform distribution over the sphere $\\|v\\|^2=d$.  To avoid the confusion, we have removed our inaccurate statement regarding the Gaussian smoothing in our text.
>
> * **Q2**: With which distribution do you sample for the estimator of the gradient used in DAP?
>
>   **A**: We simply use the uniform distribution over sphere to obtain the gradient estimation. The choice of this distribution can be set as a hyper-parameter, while it is not a crucial factor in the final performance.
>
> **Bibliography**: We appreciate the reviewer providing these highly relevant references. We have updated the bibliography and revised the introduction section to include these results.
>
> **Minor comments**: Thanks for the careful reading. We have thoroughly revised the paper to fix typos, clarified the needs of requiring $\mu \to 0$ for obtaining the unbiasedness, added discussion on the error caused by the Taylor formula approximation, highlighted that the DAP is not directly available, and improved the overall clarity.

---

> > ### Comment · Reviewer_6hyB · 2024-11-26
> >
> > Ok for me

---

### Meta-Review · Area_Chair_tACs · 2024-12-20

**Metareview:**

This paper investigates the two-point zeroth-order gradient estimator, focusing on optimizing random perturbation distributions to minimize variance. The authors identify two optimal strategies: fixed-length perturbations and the novel directionally aligned perturbations (DAPs), which enhance accuracy along critical directions. They provide theoretical foundations, including Theorem 2.2, and a practical sampling strategy for DAPs. Empirical results on synthetic and language model optimization tasks demonstrate DAPs' effectiveness, outperforming traditional methods under specific conditions.

This paper addresses a problem of interest to the optimization community. The method proposed is novel and has nice theoretical grounding and empirical validation. The paper also establishes that stochastic gradient descent with two-point gradient estimation achieves the best-known sample complexity when using minimum-variance perturbations.

Although the review team give an overall positive review, experimental validation is the main weakness. The improvements demonstrated by DAPs over other perturbation methods are marginal and lack robust comparisons against alternative methods like random coordinate sampling or the Rademacher distribution. Additionally, the experiments focus on a single practical application, limiting the generality of the findings. Lastly, the abstract claims to identify the optimal distribution, which is misleading. The paper instead provides sufficient conditions for optimality under specific assumptions, and this wording should be adjusted.

**Additional Comments On Reviewer Discussion:**

All reviewers believe this paper makes significant contribution to the literature on zero-order methods

Reviewer 6hyB and Reviewer BjSg mainly asked for some clarifications and the authors answered in details.

Reviewer F997's main concern was that the theory in this paper cannot explain the reason why Gaussian or some other distributions can achieve fast convergence rate. However, he/she believed this work still has its value and raised the score after rebuttal.

Reviewer gWHS had questions on the "only if " part of theorem 2.2 and the numerical experiments. The authors addressed them well and the reviewer then raised the score.

Reviewer aCKy requested the authors claimed their contributions more clearly using more precise words. Also, the reviewer questioned the experimental validation, pointing out that the improvement is small. The authors pointed out additional experiments where the proposed method lead to big improvements. The reviewer then increased his/her score.

---

### Decision · Program_Chairs · 2025-01-22

Accept (Spotlight)